# Continuized Acceleration for Quasar Convex Functions in Non-Convex Optimization

**Jun-Kun Wang and Andre Wibisono**
Department of Computer Science, Yale University
{jun-kun.wang,andre.wibisono}@yale.edu

## Abstract

Quasar convexity is a condition that allows some first-order methods to efficiently minimize a function even when the optimization landscape is non-convex. Previous works develop near-optimal accelerated algorithms for minimizing this class of functions, however, they require a subroutine of binary search which results in multiple calls to gradient evaluations in each iteration, and consequently the total number of gradient evaluations does not match a known lower bound. In this work, we show that a recently proposed continuized Nesterov acceleration can be applied to minimizing quasar convex functions and achieves the optimal bound with a high probability. Furthermore, we find that the objective functions of training generalized linear models (GLMs) satisfy quasar convexity, which broadens the applicability of the relevant algorithms, while known practical examples of quasar convexity in non-convex learning are sparse in the literature. We also show that if a smooth and one-point strongly convex, Polyak-Łojasiewicz, or quadratic-growth function satisfies quasar convexity, then attaining an accelerated linear rate for minimizing the function is possible under certain conditions, while acceleration is not known in general for these classes of functions.

## 1 Introduction

Momentum has been the main workhorse for training machine learning models (Kingma & Ba, 2015; Wilson et al., 2017; Loshchilov & Hutter, 2019; Reddi et al., 2018; He et al., 2016; Simonyan & Zisserman, 2015; Krizhevsky et al., 2012). In convex learning and optimization, several momentum methods have been developed under different machineries, which include the ones built on Nesterov's estimate sequence (Nesterov, 1983; 2013), methods derived from ordinary differential equations and continuous-time techniques, (Krichene et al., 2015; Scieur et al., 2017; Attouch et al., 2018; Su et al., 2014; Wibisono et al., 2016; Shi et al., 2018; Diakonikolas & Orecchia, 2019), approaches based on dynamical systems and control (Hu & Lessard, 2017; Wilson et al., 2021), algorithms generated from playing a two-player zero-sum game via no-regret learning strategies (Wang et al., 2021a; Wang & Abernethy, 2018; Cohen et al., 2021), and a recently introduced continuized acceleration (Even et al., 2021). On the other hand, in the non-convex world, despite numerous empirical evidence confirms that momentum methods converge faster than gradient descent (GD) in several applications, see e.g., Sutskever et al. (2013); Leclerc & Madry (2020), first-order accelerated methods that provably find a global optimal point are sparse in the literature. Indeed, there are just few results showing acceleration over GD that we are aware. Wang et al. (2021b) show Heavy Ball has an accelerated linear rate for training an over-parametrized ReLU network and a deep linear network, where the accelerated linear rate has a square root dependency on the condition number of a neural tangent kernel matrix at initialization, while the linear rate of GD depends linearly on the condition number. A follow-up work of Wang et al. (2022) shows that Heavy Ball has an acceleration for minimizing a class of Polyak-Łojasiewicz functions (Polyak, 1963). When the goal is not finding a global optimal point but a first-order stationary point, some benefits of incorporating the dynamic of momentum can be shown (Cutkosky & Orabona, 2019; Cutkosky & Mehta, 2021; Levy et al., 2021). Nevertheless, theoretical-grounded momentum methods in non-convex optimization are still less investigated to our knowledge.

With the goal of advancing the progress of momentum methods in non-convex optimization in mind, we study efficiently solving $\min_w f(w)$, where the function $f(\cdot)$ satisfies *quasar-convexity* (Hinder

et al., 2020; Hardt et al., 2018; Nesterov et al., 2019; Guminov & Gasnikov, 2017; Bu & Mesbahi, 2020), which is defined in the following. Under quasar convexity, it can be shown that GD or certain momentum methods can globally minimize a function even when the optimization landscape is non-convex.

**Definition 1.** *(**Quasar convexity**) Let $\rho > 0$. Denote $w_*$ a global minimizer of $f(\cdot) : \mathbb{R}^d \to \mathbb{R}$. The function $f(\cdot)$ is $\rho$-quasar convex if for all $w \in \mathbb{R}^d$, one has:*

$$f(w_*) \geq f(w) + \frac{1}{\rho}\langle \nabla f(w), w_* - w \rangle. \tag{1}$$

*For $\mu > 0$, the function $f(\cdot)$ is $(\rho, \mu)$-strongly quasar convex if for all $w \in \mathbb{R}^d$, one has:*

$$f(w_*) \geq f(w) + \frac{1}{\rho}\langle \nabla f(w), w_* - w \rangle + \frac{\mu}{2}\|w_* - w\|^2. \tag{2}$$

For more characterizations of quasar convexity, we refer the reader to Hinder et al. (2020) (Appendix D in the paper), where a thorough discussion is provided. Recall that a function $f(\cdot)$ is $L$-smooth if $f(x) \leq f(y) + \langle \nabla f(y), x - y \rangle + \frac{L}{2}\|x - y\|^2$ for any $x$ and $y$, where $L > 0$ is the smoothness constant. For minimizing $L$-smooth and $\rho$-quasar convex functions, the algorithm of Hinder et al. (2020) takes $O\left(\frac{L^{1/2}\|w_0 - w_*\|}{\rho\epsilon^{1/2}}\right)$ number of iterations and $O\left(\frac{L^{1/2}\|w_0 - w_*\|}{\rho\epsilon^{1/2}}\log\left(\frac{1}{\rho\epsilon}\right)\right)$ total number of function and gradient evaluations for getting an $\epsilon$-optimality gap. For $L$-smooth and $(\rho, \mu)$-strongly quasar convex functions, the algorithm of Hinder et al. (2020) takes $O\left(\frac{\sqrt{L/\mu}}{\rho}\log\left(\frac{V}{\epsilon}\right)\right)$ number of

iterations and $O\left(\frac{\sqrt{L/\mu}}{\rho}\log\left(\frac{V}{\epsilon}\right)\log\left(\frac{L/\mu}{\rho}\right)\right)$ number of function and gradient evaluations, where $V := f(w_0) - f(w_*) + \frac{\mu}{2}\|z_0 - w_*\|^2$, and $w_0$ and $z_0$ are some initial points. Both results of Hinder et al. (2020) improve those in the previous works of Nesterov et al. (2019) and Guminov & Gasnikov (2017) for minimizing quasar and strongly quasar convex functions. A lower bound $\Omega\left(\frac{L^{1/2}\|w_0 - w_*\|}{\rho\epsilon^{1/2}}\right)$ on the number of gradient evaluations for minimizing quasar convex functions via any first-order deterministic methods is also established in Hinder et al. (2020). The additional logarithmic factors in the (upper bounds of the) number of gradient evaluations, compared to the iteration complexity, result from a binary-search subroutine that is executed in each iteration to determine the value of a specific parameter of the algorithm. A similar concern applies to Bu & Mesbahi (2020), where the algorithm assumes an oracle is available but its implementation needs a subroutine which demands multiple function and gradient evaluations in each iteration. Hence, the open questions are whether the additional logarithmic factors in the total number of gradient evaluations can be removed and whether function evaluations are necessary for an accelerated method to minimize quasar convex functions.

We answer them by showing an accelerated *randomized* algorithm that avoids the subroutine, makes only one gradient call per iteration, and does not need function evaluations. Consequently, the complexity of gradient calls does not incur the additional logarithmic factors as the previous works, and, perhaps more importantly, the computational cost per iteration is significantly reduced. The proposed algorithms are built on the *continuized discretization* technique that is recently introduced by Even et al. (2021) to the optimization community, which offers a nice way to implement a continuous-time dynamic as a discrete-time algorithm. Specifically, the technique allows one to use differential calculus to design and analyze an algorithm in continuous time, while the discretization of the continuized process does not suffer any discretization error thanks to the fact that the Poisson process can be simulated exactly. Our acceleration results in this paper champion the approach, and provably showcase the advantage of momentum over GD for minimizing quasar convex functions.

While previous works of quasar convexity are theoretically interesting, a lingering issue is that few examples are known in non-convex machine learning. While some synthetic functions are shown in previous works (Hinder et al., 2020; Nesterov et al., 2019; Guminov & Gasnikov, 2017), the only practical non-convex learning applications that we are aware are given by Hardt et al. (2018), where they show that for learning a class of linear dynamical systems, a relevant objective function over a convex constraint set satisfies quasar convexity, and by Foster et al. (2018), where they show that a robust linear regression with Tukey's biweight loss and a GLM with an increasing link function satisfy quasar convexity, under the assumption that the link function has a bounded

second derivative (which excludes the case of Leaky-ReLU). In this work, we find that the objective functions of learning GLMs with link functions being logistic, quadratic, ReLU, or Leaky-ReLU satisfy (strong) quasar convexity, under mild assumptions on the data distribution. We also establish connections between strong quasar convexity and one-point convexity (Guille-Escuret et al., 2022; Kleinberg et al., 2018), the Polyak-Łojasiewicz (PL) condition (Polyak, 1963; Karimi et al., 2016), and the quadratic-growth (QG) condition (Drusvyatskiy & Lewis, 2018). Our findings suggest that investigating minimizing quasar convex functions is not only theoretically interesting, but is also practical for certain non-convex learning applications.

To summarize, our contributions include:

- For minimizing functions satisfying quasar convexity or strong quasar convexity, we show that the continuized Nesterov acceleration not only has the optimal iteration complexity, but also makes the same number of gradient calls required to get an expected $\epsilon$-optimality gap or an $\epsilon$-gap with high probability. The continuized Nesterov acceleration avoids multiple gradient calls in each iteration, in contrast to the previous works. We also propose an accelerated algorithm that uses stochastic pseudo-gradients for learning a class of GLMs.
- We find that GLMs with various link functions satisfy quasar convexity. Moreover, we show that if a smooth one-point convex, PL, or QG function satisfies quasar convexity, then acceleration for minimizing the function is possible under certain conditions, while acceleration over GD is not known for these classes of functions in general in the literature.

## 2    PRELIMINARIES

**Related works of gradient-based algorithms for structured non-convex optimization:**
Studying gradient-based algorithms under some relaxed notions of convexity has seen a growing interest in non-convex optimization, e.g., (Gower et al., 2021; Vaswani et al., 2019; 2022; Jin, 2020). These variegated notions include one-point convexity (Guille-Escuret et al., 2022; Kleinberg et al., 2018), the PL condition (Polyak, 1963; Karimi et al., 2016), the QG condition (Drusvyatskiy & Lewis, 2018), the error bound condition (Luo & Tseng, 1993; Drusvyatskiy & Lewis, 2018), local quasi convexity (Hazan et al., 2016), the regularity condition (Chi et al., 2019), variational coherence (Zhou et al., 2017), and quasar convexity (Hinder et al., 2020; Hardt et al., 2018; Nesterov et al., 2019; Guminov & Gasnikov, 2017; Bu & Mesbahi, 2020). For more details, we refer the reader to the references therein.

**The continuized technique of designing optimization algorithms:**
The continuized technique was introduced in Aldous & Fill (2002) under the subject of Markov chain and was recently used in optimization by Even et al. (2021), where they consider the following random process and build a connection to Nesterov's acceleration (Nesterov, 1983; 2013):

$$dw_t = \eta_t(z_t - w_t)dt - \gamma_t \nabla f(w_t)dN(t)$$
$$dz_t = \eta'_t(w_t - z_t)dt - \gamma'_t \nabla f(w_t)dN(t), \tag{3}$$

in which $\eta_t, \eta'_t, \gamma_t, \gamma'_t$ are parameters to be chosen and $dN(t)$ is the Poisson point measure. More precisely, one has $dN(t) = \sum_{k \geq 1} \delta_{T_k}(dt)$, where the random times $T_1, T_2, \ldots, T_k, \ldots$ are such that the increments $T_1, T_2 - T_1, T_3 - T_2, \ldots$ follow i.i.d. from the exponential distribution with mean 1 (so $\mathbb{E}[T_k] = k$). Between the random times, the continuized process (3) reduces to a system of ordinary differential equations:

$$dw_t = \eta_t(z_t - w_t)dt \tag{4}$$
$$dz_t = \eta'_t(w_t - z_t)dt. \tag{5}$$

At the random time $T_k$, the dynamic (3) is equivalent to taking GD steps:

$$w_{T_k} = w_{T_{k-}} - \gamma_{T_k} \nabla f(w_{T_{k-}}) \tag{6}$$
$$z_{T_k} = z_{T_{k-}} - \gamma'_{T_k} \nabla f(w_{T_{k-}}). \tag{7}$$

A nice feature of this continuized technique is that one can implement the dynamic (3) without causing any discretization error, thanks to the fact that the Poisson process can be simulated exactly. In contrast, other continuous-time approaches (Krichene et al., 2015; Scieur et al., 2017; Attouch

et al., 2018; Su et al., 2014; Wibisono et al., 2016; Shi et al., 2018; Diakonikolas & Orecchia, 2019) do not enjoy such a benefit. The formal statement of the *continuized discretization* is replicated as follows.

**Lemma 1.** *(Theorem 3 in Even et al. (2021)) The discretization of the continuized Nesterov acceleration (3) can be implemented as $\tilde{w}_k := w_{T_k}$, $\tilde{v}_k := w_{T_{k+1}-}$, $\tilde{z}_k := z_{T_k}$. Furthermore, the update of the discretized process is in the following form:*

$$\tilde{v}_k = \tilde{w}_k + \tau_k(\tilde{z}_k - \tilde{w}_k) \tag{8}$$

$$\tilde{w}_{k+1} = \tilde{v}_k - \tilde{\gamma}_{k+1}\nabla f(\tilde{v}_k) \tag{9}$$

$$\tilde{z}_{k+1} = \tilde{z}_k + \tau'_k(\tilde{v}_k - \tilde{z}_k) - \tilde{\gamma}'_{k+1}\nabla f(\tilde{v}_k), \tag{10}$$

*where $\tau_k, \tau'_k, \tilde{\gamma}_k, \tilde{\gamma}'_k$ are random parameters that are functions of $\eta_t, \eta'_t, \gamma_t$, and $\gamma'_t$.*

We replicate the proof of Lemma 1 in Appendix B. Using the continuized technique, Even et al. (2021) analyze the continuized Nesterov acceleration (3) for minimizing smooth convex functions and smooth strongly convex functions with an application in asynchronous distributed optimization.

## 3 MAIN RESULTS: APPLICATION ASPECTS

### 3.1 EXAMPLES OF QUASAR CONVEXITY

We start by identifying a class of functions that satisfy quasar convexity. To get the ball rolling, we need to introduce two notions first.

**Definition 2.** *($C_v$-generalized variational coherence w.r.t. a function $h(\cdot, \cdot)$) Denote $w_* \in \mathbb{R}^d$ a global minimizer of a function $f(\cdot)$. We say that the function $f(\cdot)$ is generalized variational coherent with the parameter $C_v > 0$ if for all $w \in \mathbb{R}^d$, one has: $\langle \nabla f(w), w - w_* \rangle \geq C_v h(w, w_*)$, where $h(w, w_*) : \mathbb{R}^d \times \mathbb{R}^d \to \mathbb{R}_+$ is a non-negative function whose inputs are $w$ and $w_*$.*

Observe that if a function is generalized variational coherent, then it is variational coherent, i.e., $\langle \nabla f(w), w - w_* \rangle \geq 0$, which is a condition that allows an almost-sure convergence to $w_*$ via mirror descent (Zhou et al., 2017). Also, when the non-negative function $h(w, w_*)$ is a squared $l_2$ norm, i.e., $h(w, w_*) = \|w - w_*\|_2^2$, it becomes one-point convexity, i.e., $\langle \nabla f(w), w - w_* \rangle \geq C_v \|w - w_*\|_2^2$. In the literature, a few non-convex learning problems have been shown to exhibit one-point convexity, see e.g., Yehudai & Shamir (2020); Sattar & Oymak (2022); Li & Yuan (2017); Kleinberg et al. (2018). However, Guille-Escuret et al. (2022) recently show that for minimizing the class of functions that are one-point convex w.r.t. a global minimizer $w_*$ and have gradient Lipschitzness in the sense that $\|\nabla f(w) - \nabla f(w_*)\|_2 \leq L\|w - w_*\|_2$ for any $w \in \mathbb{R}^d$ (which is called the upper error bound condition in their terminology), GD is optimal among *any* first-order methods, which suggests that a different condition than the upper error bound condition might be necessary to show acceleration over GD for functions satisfying one-point convexity.

**Definition 3.** *($C_l$-generalized smoothness w.r.t. a function $h(\cdot, \cdot)$) Denote $w_* \in \mathbb{R}^d$ a global minimizer of a function $f(\cdot)$. We say that the function $f(\cdot)$ is generalized smooth with the parameter $C_l > 0$ if for all $w \in \mathbb{R}^d$, one has: $f(w) - f(w_*) \leq C_l h(w, w_*)$, where $h(w, w_*) : \mathbb{R}^d \times \mathbb{R}^d \to \mathbb{R}_+$ is a non-negative function whose inputs are $w$ and $w_*$.*

We see that if a function $f(\cdot)$ is $L$-smooth w.r.t. a norm $\|\cdot\|$, then it is $\frac{L}{2}$-generalized smooth w.r.t. the square norm, i.e., $h(w, w_*) = \|w - w_*\|^2$.

**Lemma 2.** *If $f(\cdot)$ is $C_v$-generalized variational coherent and $C_l$-generalized smooth w.r.t. the same non-negative function $h(\cdot, \cdot)$, then the function satisfies $\rho$-quasar convexity with $\rho = \frac{C_v}{C_l}$.*

*Proof.* Using the definitions, we have $f(w) - f(w_*) \leq C_l h(w, w_*) \leq \frac{C_l}{C_v}\langle \nabla f(w), w - w_* \rangle$. □

Lemma 2 could be viewed as a modified result of Lemma 5 in Foster et al. (2018), where the authors show that a GLM with the link function having a bounded second derivative and a positive first derivative satisfies quasar convexity. In the following, we provide three more examples of quasar convexity, while the proofs are deferred to Appendix C. For these examples, we assume that each

sample $x \in \mathbb{R}^d$ is i.i.d. from a distribution $\mathcal{D}$, and that there exists a $w_* \in \mathbb{R}^d$ such that its label is generated as $y = \sigma(w_*^\top x)$, where $\sigma(\cdot) : \mathbb{R} \to \mathbb{R}$ is the link function of a GLM. We consider minimizing the square loss function:

$$f(w) := \mathbb{E}_{x \sim \mathcal{D}} \left[ \tfrac{1}{2} \left( \sigma(w^\top x) - y \right)^2 \right]. \tag{11}$$

### 3.1.1 EXAMPLE 1: (GLMS WITH INCREASING LINK FUNCTIONS)

**Lemma 3.** *Suppose that the link function $\sigma(z)$ is $L_0$-Lipschitz and $\alpha$-increasing, i.e., $\sigma'(z) \geq \alpha > 0$ for all $z > \mathbb{R}$. Then, the loss function (11) is $\alpha^2$-generalized variational coherent and $\frac{L_0^2}{2}$-generalized smooth w.r.t. $h(w, w_*) = \mathbb{E}_{x \sim \mathcal{D}} \left[ ((w - w_*)^\top x)^2 \right]$. Therefore, it is $\rho = \frac{2\alpha^2}{L_0^2}$-quasar convex.*

An example of the link functions that satisfy the assumption is the Leaky-ReLU, i.e., $\sigma(z) = \max(\alpha z, z)$, where $\alpha > 0$. If we further assume that the models $w, w_*$, and the features $x$ in (11) have finite length so that the input to the link function $\sigma(\cdot)$ is bounded, then the logistic link function, i.e., $\sigma(z) = (1 + \exp(-z))^{-1}$, is another example.

### 3.1.2 EXAMPLE 2: (PHASE RETRIEVAL)

When the link function is quadratic, i.e., $\sigma(z) = z^2$, the objective function becomes that of phase retrieval, see e.g., Yonel & Yazici (2020); Chi et al. (2019). White et al. (2016), Yonel & Yazici (2020) show that in the neighborhood of the global minimizers $\pm w_*$, the function satisfies one-point convexity in terms of the $l_2$ norm when the data distribution $\mathcal{D}$ follows a Gaussian distribution, for which a specialized initialization technique called the spectral initialization finds a point in the neighborhood (Ma et al., 2020). As discussed earlier, one-point convexity is equivalent to generalized coherence w.r.t. the square norm, i.e., $h(w, w_*) = \|w - w_*\|_2^2$. Therefore, by Lemma 2, to show quasar convexity for all $w$ in the neighborhood of $\pm w_*$, it remains to show that the objective function is generalized smooth w.r.t. the square norm $\|w - w_*\|_2^2$.

**Lemma 4.** *Assume that there exists a finite constant $C_R > 0$ such that all $w \in \mathbb{R}^d$ in the balls of radius $R$ centered at $\pm w_*$ satisfy $\mathbb{E}_{x \sim D} \left[ \left( (w + w_*)^\top x \right)^2 \|x\|_2^2 \right] \leq C_R$. Then, the loss function (11) is $\frac{1}{2} C_R$-generalized smooth w.r.t. $h(w, w_*) = \|w - w_*\|_2^2$.*

An example of the distribution $\mathcal{D}$ that satisfies the assumption in Lemma 4 is a Gaussian distribution.

### 3.1.3 EXAMPLE 3: (LEARNING A SINGLE RELU)

When the link function is ReLU, i.e., $\sigma(z) = \max\{0, z\}$, Theorem 4.2 in Yehudai & Shamir (2020) shows that under mild assumptions of the data distribution, e.g., $\mathcal{D}$ is a Gaussian, the objective function is one-point convex in terms of the $l_2$ norm. Therefore, as the case of phase retrieval, it remains to show generalized smoothness w.r.t. the square norm $\|w - w_*\|_2^2$ for showing quasar convexity.

**Lemma 5.** *When the link function is ReLU, the loss function (11) is $\frac{1}{2} \mathbb{E}_{x \sim D}[\|x\|_2^2]$-generalized smooth w.r.t. $h(w, w_*) = \|w - w_*\|_2^2$.*

## 3.2 EXAMPLES OF STRONG QUASAR CONVEXITY

In this subsection, we switch to investigate strong quasar convexity. We establish its connections to one-point convexity, the PL condition, and the QG condition.

### 3.2.1 ONE-POINT CONVEX FUNCTIONS WITH QUASAR CONVEXITY

It turns out that if a $C_v$-one-point convex function $f(\cdot)$ satisfies $\rho$-quasar convexity, then it also satisfies strong quasar convexity. Specifically, we have the following lemma.

**Lemma 6.** *Suppose that the function $f(\cdot)$ satisfies $C_v$-one-point convexity and $\hat{\rho}$-quasar convexity. Then, it is also $\left( \rho = \frac{\hat{\rho}}{\theta}, \mu = \frac{2C_v(\theta - 1)}{\hat{\rho}} \right)$-strongly quasar convex for any $\theta > 1$.*

The proof is deferred to Appendix C.4. By Lemma 6, phase retrieval and ReLU regression illustrated in the previous subsection can also be strongly quasar convex.

### 3.2.2 POLYAK-ŁOJASIEWICZ (PL) OR QUADRATIC-GROWTH (QG) FUNCTIONS WITH QUASAR CONVEXITY

Recall that a function $f(\cdot)$ satisfies $\nu$-QG w.r.t. a global minimizer $w_* \in \mathbb{R}^d$ if $f(w) - f(w_*) \geq \frac{\nu}{2}\|w - w_*\|^2$ for some $\nu > 0$ and all $w \in \mathbb{R}^d$ (Drusvyatskiy & Lewis, 2018; Karimi et al., 2016). Recall also that a function $f(\cdot)$ satisfies $\nu$-PL if $2\nu(f(w) - f(w_*)) \leq \|\nabla f(w)\|^2$ for some $\nu > 0$ and all $w \in \mathbb{R}^d$ (Karimi et al., 2016). It is known that a $\nu$-PL function satisfies $\nu$-QG, see e.g., Appendix A in Karimi et al. (2016). The notion of PL has been discovered in various non-convex problems recently (Altschuler et al., 2021; Oymak & Soltanolkotabi, 2019; Chizat, 2021; Merigot et al., 2021). We show in Lemma 7 below that if a $\nu$-QG function $f(\cdot)$ satisfies quasar convexity, then it also satisfies strong quasar convexity.

**Lemma 7.** *Suppose that the function $f(\cdot)$ is $\nu$-QG and $\hat{\rho}$-quasar convex w.r.t. a global minimizer $w_*$. Then, it is also $(\rho = \hat{\rho}\theta, \mu = \frac{\nu(1-\theta)}{\theta})$-strongly quasar convex for any $\theta < 1$.*

Lemma 8 in the following shows that GLMs with increasing link functions satisfy QG under certain distributions $\mathcal{D}$, e.g., Gaussian, and hence they are strongly quasar convex by Lemma 7 and Lemma 3.

**Lemma 8.** *Following the setting of Lemma 3, assume that the smallest eigenvalue of the matrix $\mathbb{E}_{x\sim\mathcal{D}}[xx^\top]$ satisfies $\lambda_{\min}(\mathbb{E}_{x\sim\mathcal{D}}[xx^\top]) > 0$. Then, the function (11) is $\alpha^2 \lambda_{\min}(\mathbb{E}_{x\sim\mathcal{D}}[xx^\top])$-QG.*

The proofs of Lemma 7 and Lemma 8 are available in Appendix C.5.

## 4 MAIN RESULTS: ALGORITHMIC ASPECTS

We first analyze the continuized Nesterov acceleration (3) and its discrete-time version (8)-(10) for minimizing quasar convex functions.

**Theorem 1.** *Assume that the function $f(\cdot)$ is $L$-smooth and $\rho$-quasar convex. Let $\eta_t = \frac{2}{\rho t}, \eta'_t = 0, \gamma_t = \frac{1}{L}$, and $\gamma'_t = \frac{\rho t}{2L}$. Then, the update $w_t$ of the continuized algorithm (3) satisfies*

$$\mathbb{E}[f(w_t) - f(w_*)] \leq \frac{2L\|z_0 - w_*\|^2}{\rho^2 t^2}.$$

*Furthermore, for the update $\tilde{w}_k$ of the discrete-time algorithm (8)-(10), if the parameters are chosen as $\tau_k = 1 - \left(\frac{T_k}{T_{k+1}}\right)^{2/\rho}, \tau'_k = 0, \tilde{\gamma}_k = \frac{1}{L}$, and $\tilde{\gamma}'_k = \frac{\rho T_k}{2L}$, then*

$$\mathbb{E}\left[T_k^2 \left(f(\tilde{w}_k) - f(w_*)\right)\right] \leq \frac{2L\|\tilde{z}_0 - w_*\|^2}{\rho^2}.$$

It is noted that the expectation $\mathbb{E}$ is with respect to the Poisson process, which is the only source of randomness in the continuized Nesterov acceleration. By applying some concentration inequalities, we can get a bound on the optimal gap with a high probability from Theorem 1.

**Corollary 1.** *The update $\tilde{w}_k$ of the algorithm (8)-(10) with the same parameters indicated in Theorem 1 satisfies $f(\tilde{w}_k) - f(w_*) \leq \frac{2c_0 L\|\tilde{z}_0 - w_*\|^2}{(1-c)^2 \rho^2 k^2}$, with probability at least $1 - \frac{1}{c^2 k} - \frac{1}{c_0}$ for any $c \in (0, 1)$ and $c_0 > 1$.*

Corollary 1 implies that $K = O\left(\frac{L^{1/2}\|\tilde{z}_0 - w_*\|}{\rho \epsilon^{1/2}}\right)$ number of gradient calls is sufficient for the discrete-time algorithm to get an $\epsilon$-optimality gap with a high probability, since the discrete-time algorithm only queries one gradient in each iteration $k$.

Next we analyze the convergence rate for minimizing $(\rho, \mu)$-strongly quasar-convex functions.

**Theorem 2.** *Assume that the function $f(\cdot)$ is $L$-smooth and $(\rho, \mu)$-strongly quasar convex, where $\mu > 0$. Let $\gamma_t = \frac{1}{L}, \gamma'_t = \frac{1}{\sqrt{\mu L}}, \eta'_t = \rho\sqrt{\frac{\mu}{L}}$, and $\eta_t = \sqrt{\frac{\mu}{L}}$. Then, the update $w_t$ of the continuized algorithm (3) satisfies*

$$\mathbb{E}[f(w_t) - f(w_*)] \leq \left(f(w_0) - f(w_*) + \frac{\mu}{2}\|z_0 - w_*\|^2\right) \exp\left(-\rho\sqrt{\frac{\mu}{L}}t\right).$$

*Furthermore, for the update $\tilde{w}_k$ of the discrete-time algorithm (8)-(10), if the parameters are chosen as $\tau_k = \frac{1}{1+\rho}\left(1 - \exp\left(-(1+\rho)\sqrt{\frac{\mu}{L}}(T_{k+1} - T_k)\right)\right), \tau'_k = \frac{\rho\left(1-\exp\left(-(1+\rho)\sqrt{\frac{\mu}{L}}(T_{k+1} - T_k)\right)\right)}{\rho+\exp\left(-(1+\rho)\sqrt{\frac{\mu}{L}}(T_{k+1} - T_k)\right)}, \tilde{\gamma}_k = \frac{1}{L}$, and $\tilde{\gamma}'_k = \frac{1}{\sqrt{\mu L}}$, then*

$$\mathbb{E}\left[\exp\left(\rho\sqrt{\frac{\mu}{L}}T_k\right)\left(f(\tilde{w}_k) - f(w_*)\right)\right] \leq f(\tilde{w}_0) - f(w_*) + \frac{\mu}{2}\|\tilde{z}_0 - w_*\|^2.$$

**Corollary 2.** *The update $\tilde{w}_k$ of the algorithm (8)-(10) with the same parameters indicated in Theorem 2 satisfies $f(\tilde{w}_k) - f(w_*) \leq c_0 \left( f(\tilde{w}_0) - f(w_*) + \frac{\mu}{2}\|\tilde{z}_0 - w_*\|^2 \right) \exp\left(-\rho\sqrt{\frac{\mu}{L}}(1-c)k\right)$, with probability at least $1 - \frac{1}{c^2 k} - \frac{1}{c_0}$ for any $c \in (0,1)$ and $c_0 > 1$.*

The proof of the above theorems and corollaries are available in Appendix D. Denote $V := f(\tilde{w}_0) - f(w_*) + \frac{\mu}{2}\|\tilde{z}_0 - w_*\|^2$. Theorem 2 and Corollary 2 show that the proposed algorithm takes $O\left(\frac{\sqrt{L/\mu}}{\rho} \log\left(\frac{V}{\epsilon}\right)\right)$ number of iterations with the same number of gradient evaluations to get an $\epsilon$-expected optimality gap and an $\epsilon$-optimality gap with a high probability respectively. Together with Theorem 1 and Corollary 1, these theoretical results show that the continuized Nesterov acceleration has an advantage compared to the existing algorithms of minimizing quasar and strongly quasar-convex functions (Hinder et al., 2020; Bu & Mesbahi, 2020; Nesterov et al., 2019; Guminov & Gasnikov, 2017), as it avoids multiple gradient calls in each iteration and does not need function evaluations to have an $\epsilon$-gap with a high probability. On the other hand, it should be emphasized that the guarantees in the aforementioned works are deterministic bounds, while ours is an expected one or a high-probability bound.

Recall Lemma 2 suggests that $C_v$-one-point convexity and $L$-smoothness implies $\rho = \frac{2C_v}{L}$ quasar convexity. Furthermore, Lemma 6 states that $\hat{\rho}$ quasar convexity and $C_v$-one-point convexity actually implies $\left(\rho = \frac{\hat{\rho}}{\theta}, \mu = \frac{2C_v(\theta-1)}{\hat{\rho}}\right)$-strongly quasar convex for any $\theta > 1$. By combining Lemma 2 and Lemma 6, we find that $C_v$-one-point convexity and $L$-smoothness implies $(\rho = \frac{2C_v}{L\theta}, \mu = L(\theta-1))$-strong quasar convexity, for any $\theta > 1$. By substituting $(\rho = \frac{2C_v}{L\theta}, \mu = L(\theta-1))$ into the complexity $O\left(\frac{\sqrt{L/\mu}}{\rho} \log\left(\frac{V}{\epsilon}\right)\right)$ indicated by Corollary 2, we see that the required number of iterations to get an $\epsilon$ gap with high probability for minimizing functions that satisfy $C_v$-one-point convexity and $L$-smoothness via the proposed algorithm is $O\left(\frac{L}{C_v}\frac{\theta}{\sqrt{\theta-1}} \log\left(\frac{V}{\epsilon}\right)\right) = O\left(\frac{L}{C_v} \log\left(\frac{V}{\epsilon}\right)\right)$, where we simply let $\theta = 2$. On the other hand, Guille-Escuret et al. (2022) consider minimizing a class of functions that satisfies $C_v$-one-point convexity and a condition called the $L$-upper error bound condition ($L$-EB$^+$). A function satisfies $L$-EB$^+$ if $\|\nabla f(w) - \nabla f(w_*)\|_2 \leq L\|w - w_*\|_2$ for a fixed minimizer $w_*$ and any $w \in \mathbb{R}^d$. Guille-Escuret et al. (2022) show that the optimal iteration complexity $k$ to have $\frac{\|w_k - w_*\|_2^2}{\|w_0 - w_*\|_2^2} = \hat{\epsilon}$ for minimizing the class of functions via any first-order algorithm is $k = \Theta\left(\left(\frac{L}{C_v}\right)^2 \log\left(\frac{1}{\hat{\epsilon}}\right)\right)$ and that the optimal complexity is simply attained by GD. Our result does not contradict to this lower bound result, because $L$-smoothness and $L$-EB$^+$ are different. $L$-EB$^+$ is about gradient Lipschitzness between a minimizer $w_*$ and any $w$, not for any pair of points in $\mathbb{R}^d$, and hence does not imply $L$-smoothness. Also, $L$-smoothness does not imply $L$-EB$^+$.

Lastly, it is noted that both theorems (and corollaries) require the $L$-smoothness condition. The reader might raise a question whether the smoothness holds for the case when a GLM with the link function is ReLU or Leaky-ReLU. For this case, it has been shown that the objective (11) satisfies smoothness when the data distribution is a Gaussian distribution, e.g., Lemma 5.2 in Zhang et al. (2018).

**Continuized accelerated algorithm with stochastic pseudo-gradients for GLMs:** We also propose a *stochastic* algorithm to recover an unknown GLM $w_* \in \mathbb{R}^d$ that generates the label $y$ of a sample $x \in \mathbb{R}^d$ via $y = \sigma(w_*^\top x)$, where $\sigma(\cdot)$ is the link function. A natural metric for this task is the distance to the unknown target $w_*$, i.e., $f(w) := \frac{1}{2}\|w - w_*\|_2^2$, However, since we do not have access to $w_*$, we cannot use the gradients of $f(\cdot)$ for the update. Instead, let us consider using stochastic *pseudo-gradients*, defined as $g(w; \xi) := \left(\sigma(w^\top x) - y\right) x$, where $\xi := (x, y)$ represents a random sample drawn from the data distribution. Assume that the first derivative of the link function is positive, i.e., $\sigma'(\cdot) \geq \alpha > 0$. Then, the expectation of the dot product between the pseudo-gradient and the gradient $\nabla f(w)$ over the data distribution satisfies

$$\mathbb{E}_\xi[\langle g(w; \xi), \nabla f(w)\rangle] = \mathbb{E}_\xi[\langle \left(\sigma(w^\top x) - y\right) x, w - w_*\rangle] = \mathbb{E}_x[\langle \left(\sigma(w^\top x) - \sigma(w_*^\top x)\right) x, w - w_*\rangle]$$

$$= \mathbb{E}_x\left[\frac{\left(\sigma(w^\top x) - \sigma(w_*^\top x)\right)}{w^\top x - w_*^\top x}\left((w - w_*)^\top x\right)^2\right] \geq \alpha\mathbb{E}_x\left[\left((w - w_*)^\top x\right)^2\right],$$

$$(12)$$

which implies that taking a negative of pseudo-gradient step should make progress on minimizing the distance $\frac{1}{2}\|w - w_*\|_2^2$ on expectation when $w$ has not converged to $w_*$. That is, the update $w_{t+1} = w_t - \eta g(w_t; \xi)$ could be shown to converge to the target $w_* \in \mathbb{R}^d$ under certain conditions, where $\eta > 0$ is the step size. In fact, this algorithm is called (stochastic) GLMtron in the literature (Kakade et al., 2011). We introduce a continuized acceleration of it in the following. But before that, let us provide some necessary ingredients first.

Denote a matrix $H(w) := \mathbb{E}_x[\psi(w^\top x, w_*^\top x) x x^\top]$, where $\psi(a, b) := \frac{\sigma(a) - \sigma(b)}{a - b}$. When the data matrix satisfies $\mathbb{E}_x[x x^\top] \succeq \theta I_d$ for some $\theta > 0$ and when the derivative of the link function $\sigma(\cdot)$ satisfies $\sigma'(\cdot) \geq \alpha > 0$, one has $H(w) \succeq \mu I_d \succ 0$, where $\mu := \alpha\theta$. We assume that for any $w \in \mathbb{R}^d$, it holds that $\mathbb{E}_x \left[ \psi(w^\top x, w_*^\top x)^2 \|x\|_2^2 x x^\top \right] \preceq R^2 H(w)$ for some constant $R^2 > 0$, and also that $\mathbb{E}_x \left[ \psi(w^\top x, w_*^\top x)^2 \|x\|_{H(w)^{-1}}^2 x x^\top \right] \preceq \tilde{\kappa} H(w)$ for some constant $\tilde{\kappa} > 0$, where we denote $\|x\|_{H(w)^{-1}}^2 := x^\top H(w)^{-1} x$ and $H(w)^{-1}$ is the inverse of $H(w)$. Define $\kappa := \frac{R^2}{\mu}$. Then, we have $\tilde{\kappa} \leq \kappa$, because $\mathbb{E}_x \left[ \psi(w^\top x, w_*^\top x)^2 \|x\|_{H(w)^{-1}}^2 x x^\top \right] \preceq \frac{1}{\mu} \mathbb{E}_x \left[ \psi(w^\top x, w_*^\top x)^2 \|x\|_2^2 x x^\top \right] \preceq \frac{R^2}{\mu} H(w) = \kappa H(w)$. The assumptions can be viewed as a generalization of the assumptions made in Jain et al. (2018); Even et al. (2021) for the standard least-square regression, in which case one has $\sigma(z) = z$ and hence $\psi(\cdot, \cdot) = 1$.

Our continuized acceleration with stochastic pseudo-gradient steps can be formulated as:

$$
\begin{aligned}
dw_t &= \eta_t(z_t - w_t)dt - \gamma_t \int_\Xi g(w_t; \xi) dN(t, \xi) \\
dz_t &= \eta_t'(w_t - z_t)dt - \gamma_t' \int_\Xi g(w_t; \xi) dN(t, \xi),
\end{aligned}
\tag{13}
$$

where $\eta_t, \eta_t', \gamma_t, \gamma_t'$ are parameters, $\xi \in \Xi$ represents an i.i.d. random variable associated with a sample used to compute a stochastic pseudo-gradient $g(w; \xi)$, and $dN(t, \xi) = \Sigma_{k \geq 1} \delta_{(T_k, \xi_k)}(dt, d\xi)$ is the Poisson point measure on $\mathbb{R}_{\geq 0} \times \Xi$. We have Theorem 3 in the following, and its proof is available in Appendix E, where we also provide a convergence guarantee of the discrete-time algorithm.

**Theorem 3.** *(Continuized algorithm (13) for GLMs) Choose* $\eta_t' = \sqrt{\frac{\mu}{\tilde{\kappa}R^2}}$, $\eta_t = \sqrt{\frac{\mu}{\tilde{\kappa}R^2}}$, $\gamma_t = \frac{1}{R^2}$, *and* $\gamma_t' = \frac{1}{\sqrt{\mu\tilde{\kappa}R^2}}$. *Then, the update* $w_t$ *of (13) satisfies*

$$
\mathbb{E} \left[ \tfrac{1}{2} \|w_t - w_*\|_2^2 \right] \leq \tfrac{1}{2} \left( \|w_0 - w_*\|_2^2 + \mu \|z_0 - w_*\|_{H(w_0)^{-1}}^2 \right) \exp \left( -\sqrt{\tfrac{\mu}{\tilde{\kappa}R^2}} t \right).
$$

## 5 EXPERIMENTS

We compare the proposed continuized acceleration with GD and the accelerated method of Hinder et al. (2020) (AGD). For the method of Hinder et al. (2020), we use their implementation available online (Hinder et al., 2021). Our first set of experiments consider optimizing the empirical risks of GLMs with link functions being logistic, ReLU, and quadratic, i.e., solving $\min_w \frac{1}{n} \sum_{i=1}^n \left[ \frac{1}{2} \left( \sigma(w^\top x_i) - y_i \right)^2 \right]$, where $n$ is the number of samples. Each data point $x_i$ is sampled from the normal distribution $N(0, I_d)$ and the label $y_i$ is generated as $y_i = \sigma(w_*^\top x_i)$, where $w_* \sim N(0, I_d)$ is the true vector and $\sigma(\cdot)$ is the link function. In the experiments, we set the number of samples $n = 1000$ and the dimension $d = 50$. The initial point of all the algorithms $w_0 \in \mathbb{R}^d$ is a close-to-zero point, and is sampled as $w_0 \sim 10^{-2}\zeta$, where $\zeta \sim N(0, I_d)$. Since the continuized acceleration has randomness due to the Poisson process, it was replicated 10 runs in the experiments, and the averaged results over these runs are reported. Both the continuized acceleration and AGD of Hinder et al. (2021) need the knowledge of $L$, $\rho$, and $\mu$ for setting their parameters theoretically. We instead use the grid search and report the result under the best configuration of these parameters for each method. More precisely, we search $L$ and $\mu$ over $\{\ldots, 10^q, 5 \times 10^q, 10^{q+1}, \ldots\}$ with the constraint that $L > \mu$, where $q \in \{-2, -1, \ldots, 4\}$, and search $\rho \in \{0.01, 0.1, 0.5\}$.

Figure 1 shows the results, where we compare the performance of the algorithms in terms of the function value versus iteration, the number of gradient calls, and CPU time (seconds). From the first column of the figure, one can see that the proposed continuized acceleration is competitive with AGD of Hinder et al. (2020) in terms of the number of iterations. From the middle and the last column, the continuized acceleration shows its promising results over AGD and GD when they are

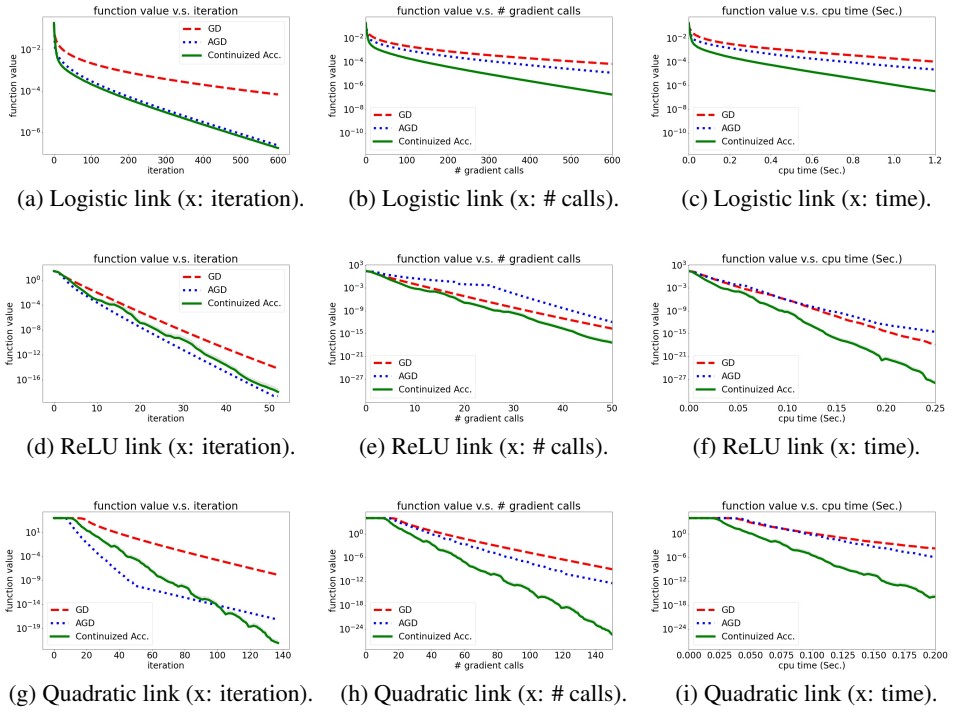

Figure 1: Comparison of the continuized Nesterov acceleration, GD, and AGD (Hinder et al., 2020).

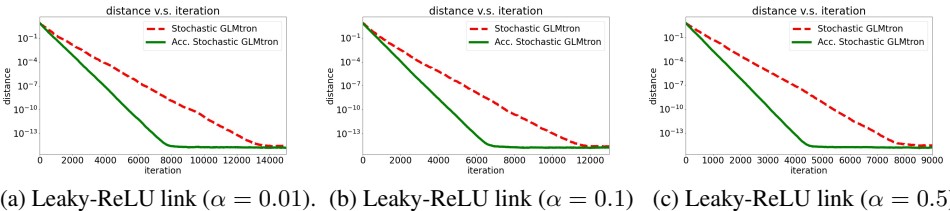

Figure 2: Distance $\|w_k - w_*\|$ v.s. iteration $k$.

measured in terms of the number of gradient calls and CPU time, which confirms that the cost of AGD per iteration is indeed higher than the continuized acceleration and showcases the advantage of the continuized acceleration. Our second set of experiments compare stochastic GLMtron and the proposed continuized acceleration of it (accelerated stochastic GLMtron), in which both algorithms randomly select a sample to compute a stochastic pseudo-gradient at each step of the update. We consider learning a GLM with a Leaky-ReLU, i.e., $\sigma(z) = \max(\alpha z, z)$ under different values of $\alpha$. Figure 2 shows the effectiveness of accelerated stochastic GLMtron, as it is significantly faster than stochastic GLMtron for recovering the true vector $w_*$.

## 6  CONCLUSION

We show that the continuized Nesterov acceleration outperforms the previous accelerated methods for minimizing quasar convex functions. Compared to the previous approaches, the continuized discretization technique provides a relatively easy way to design and analyze an accelerated algorithm for quasar convex functions. Hence, it would be interesting to check whether this technique could offer any other benefits in non-convex optimization. Specifically, can the technique help design fast algorithms for minimizing other classes of non-convex functions? On the other hand, while examples of quasar convex functions are provided in this paper, a natural question is if this property holds more broadly in modern machine learning applications. Exploring the possibilities might be another interesting direction.

ACKNOWLEDGMENTS

We thank the reviewers for constructive feedback, which helps improve the quality of this paper.

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

# A    ALGORITHMS OF HINDER ET AL. (2020)

We replicate the algorithms in Hinder et al. (2020) using our notations for the reader's reference. Their algorithms use a subroutine of binary search to determine the "mixing" parameter $\tau_k$.

---

**Algorithm 1:** AGD for $(\rho, \mu)$-strongly quasar convex function minimization in Hinder et al. (2020)

---

1: Set $\tau_k = \rho\sqrt{\frac{\mu}{L}}$, $\tilde{\gamma}_k = \frac{1}{L}$, and $\tilde{\gamma}'_k = \frac{1}{\sqrt{\mu L}}$
2: **for** $k = 0, 1, \ldots, K$ **do**
3:     $\alpha_k \leftarrow$ BINARYLINESEARCH $\left( f, w_k, z_k, b = \frac{\rho\mu}{2}, c = \sqrt{\frac{L}{\mu}}, \tilde{\epsilon} = 0 \right)$.
4:     $\tau_k \leftarrow 1 - \alpha_k$.
5:     $v_k = w_k + \tau_k(z_k - w_k)$.
6:     $w_{k+1} = v_k - \tilde{\gamma}_{k+1}\nabla f(v_k)$.
7:     $z_{k+1} = z_k + \tau'_k(v_k - z_k) - \tilde{\gamma}'_{k+1}\nabla f(v_k)$.
8: **end**
9: **return** $w_K$

---

---

**Algorithm 2:** AGD for $\rho$-quasar convex function minimization in Hinder et al. (2020)

---

1: Set $\tau_k = 0$, $\tilde{\gamma}_k = \frac{1}{L}$, and $\tilde{\gamma}'_k = \frac{\rho}{L\theta_k}$, where $\theta_k = \frac{\theta_{k-1}}{2}\left( \sqrt{(\theta_{k-1})^2 + 4} - \theta_{k-1} \right)$ for $k \geq 0$ and
   $\theta_{-1} = 1$.
2: **for** $k = 0, 1, \ldots, K$ **do**
3:     $\alpha_k \leftarrow$ BINARYLINESEARCH $\left( f, w_k, z_k, b = 0, c = \rho\left( \frac{1}{\theta_k} - 1 \right), \tilde{\epsilon} = \frac{\rho\epsilon}{2} \right)$.
4:     $\tau_k \leftarrow 1 - \alpha_k$.
5:     $v_k = w_k + \tau_k(z_k - w_k)$.
6:     $w_{k+1} = v_k - \tilde{\gamma}_{k+1}\nabla f(v_k)$.
7:     $z_{k+1} = z_k + \tau'_k(v_k - z_k) - \tilde{\gamma}'_{k+1}\nabla f(v_k)$.
8: **end**
9: **return** $w_K$

---

---

**Algorithm 3:** BINARYLINESEARCH$(f, w, z, b, c, \tilde{\epsilon}, [guess])$ (Hinder et al., 2020)

---

1: Assumptions: $f$ is $L$-smooth, $w, z \in \mathbb{R}^d$; $b, c, \tilde{\epsilon} \geq 0$; "*guess*" (optional) is in $[0, 1]$ if provided.
2: Define $g(\alpha) := f(\alpha w + (1 - \alpha)z)$ and $p := b\|w - z\|^2$.
3: **if** *guess* provided **and** $cg(guess) + guess(g'(guess) - guess \cdot p) \leq cg(1) + \tilde{\epsilon}$ **then return** *guess*;
4: **if** $g'(1) \leq \tilde{\epsilon} + p$ **then return** 1;
5: **else if** $c = 0$ or $g(0) \leq g(1) + \frac{\tilde{\epsilon}}{c}$ **then return** 0;
6: $\tau \leftarrow 1 - \frac{\tilde{\epsilon} + p}{L\|w - z\|^2}$.
7: $\textbf{lo} \leftarrow 0, \textbf{hi} \leftarrow \tau, \alpha \leftarrow \tau$.
8: **while** $cg(\alpha) + \alpha(g'(\alpha) - \alpha p) > cg(1) + \tilde{\epsilon}$ **do**
      $\alpha \leftarrow (\textbf{lo} + \textbf{hi})/2$
      **if** $g(\alpha) \leq g(\tau)$ **then**
         $\textbf{hi} \leftarrow \alpha$;
      **else**
         $\textbf{lo} \leftarrow \alpha$;
   **end**
9: **return** $\alpha$.

---

# B    PROOF OF LEMMA 1

**Lemma 1** *(Theorem 3 in Even et al. (2021)) The discretization of the continuized Nesterov accelera-tion (3) can be implemented as $\tilde{w}_k := w_{T_k}$, $\tilde{v}_k := w_{T_{k+1}-}$, $\tilde{z}_k := z_{T_k}$. Furthermore, the update of the discretized process is in the following form:*

$$\tilde{v}_k = \tilde{w}_k + \tau_k(\tilde{z}_k - \tilde{w}_k) \tag{14}$$

$$\tilde{w}_{k+1} = \tilde{v}_k - \tilde{\gamma}_{k+1}\nabla f(\tilde{v}_k) \tag{15}$$

$$\tilde{z}_{k+1} = \tilde{z}_k + \tau_k'(\tilde{v}_k - \tilde{z}_k) - \tilde{\gamma}_{k+1}'\nabla f(\tilde{v}_k), \tag{16}$$

*where $\tau_k, \tau_k', \tilde{\gamma}_k, \tilde{\gamma}_k'$ are random parameters that are functions of $\eta_t, \eta_t', \gamma_t$, and $\gamma_t'$.*

*Proof.* We replicate the proof in (Even et al., 2021) for completeness. Recall between random times, we have the ODEs

$$dw_t = \eta_t(z_t - w_t)dt \tag{17}$$

$$dz_t = \eta_t'(w_t - z_t)dt. \tag{18}$$

Integrating from $T_k$ to $T_{k+1-}$,

$$\tilde{v}_k = w_{T_{k+1}-} = w_{T_k} + \tau_k(z_{T_k} - w_{T_k}) = \tilde{w}_k + \tau_k(\tilde{z}_k - \tilde{w}_k) \tag{19}$$

$$z_{T_{k+1}-} = z_{T_k} + \tau_k''(w_{T_k} - z_{T_k}) = \tilde{z}_k + \tau_k''(\tilde{w}_k - \tilde{z}_k), \tag{20}$$

where $\tau_k$ and $\tau_k''$ depend on $\eta_t$ and $\eta_t'$ respectively. Combing the above two equations, we have

$$z_{T_{k+1}-} = \tilde{z}_k + \tau_k''\left(\frac{1}{1-\tau_k}(\tilde{v}_k - \tau_k\tilde{z}_k) - \tilde{z}_k\right) = \tilde{z}_k + \tau_k'(\tilde{v}_k - \tilde{z}_k), \tag{21}$$

where $\tau_k' := \frac{\tau_k''}{1-\tau_k}$. Furthermore, from (6) and (7), we have

$$\tilde{w}_{k+1} = w_{T_{k+1}} = w_{T_{k+1}-} - \gamma_{T_{k+1}}\nabla f(w_{T_{k+1}-}) = \tilde{v}_k - \gamma_{T_{k+1}}\nabla f(\tilde{v}_k), \tag{22}$$

$$\tilde{z}_{k+1} = z_{T_{k+1}} = z_{T_{k+1}-} - \gamma_{T_{k+1}}'\nabla f(w_{T_{k+1}-}) \overset{(21)}{=} \tilde{z}_k + \tau_k'(\tilde{v}_k - \tilde{z}_k) - \gamma_{T_{k+1}}'\nabla f(\tilde{v}_k). \tag{23}$$

Hence, $\tilde{\gamma}_{k+1} = \gamma_{T_{k+1}}$ and $\tilde{\gamma}_{k+1}' = \gamma_{T_{k+1}}'$.

$\square$

# C    MISSING PROOFS IN SECTION 3

## C.1    PROOF OF LEMMA 3

**Lemma 3** *Suppose that the link function $\sigma(z)$ is $L_0$-Lipschitz and $\alpha$-increasing, i.e., $\sigma'(z) \geq \alpha > 0$ for all $z > \mathbb{R}$. Then, the loss function (11) is $\alpha^2$-generalized variational coherent and $\frac{L_0^2}{2}$-generalized smooth w.r.t. $h(w, w_*) = \mathbb{E}_{x\sim\mathcal{D}}\left[((w - w_*)^\top x)^2\right]$. Therefore, the function (11) is $\rho = \frac{2\alpha^2}{L_0^2}$-quasar convex.*

*Proof.* We first show generalized variational coherence. We have

$$
\begin{aligned}
\langle\nabla f(w), w - w_*\rangle &\overset{(a)}{=} \mathbb{E}_{x\sim\mathcal{D}}\left[\left(\sigma(w^\top x) - \sigma(w_*^\top x)\right)\sigma'(w^\top x)\langle w - w_*, x\rangle\right] \\
&= \mathbb{E}_{x\sim\mathcal{D}}\left[\left(\frac{\sigma(w^\top x) - \sigma(w_*^\top x)}{(w - w_*)^\top x}\right)\sigma'(w^\top x)((w - w_*)^\top x)^2\right] \\
&\overset{(b)}{\geq} \alpha^2\mathbb{E}_{x\sim\mathcal{D}}\left[((w - w_*)^\top x)^2\right] \\
&= \alpha^2 h(w, w_*),
\end{aligned}
\tag{24}
$$

where (a) uses that $y = \sigma(w_*^\top x)$, (b) uses that $\frac{\sigma(w^\top x) - \sigma(w_*^\top x)}{w^\top x - w_*^\top x} \geq 0$ as $\sigma'(\cdot) \geq \alpha > 0$. Now let us switch to show generalized smoothness. We have

$$
\begin{aligned}
f(w) - f(w_*) &= \mathbb{E}_{x\sim D}\left[\frac{1}{2}\left(\sigma(w^\top x) - \sigma(w_*^\top x)\right)^2\right] \\
&\leq \frac{L_0^2}{2}\mathbb{E}_{x\sim D}\left[((w - w_*)^\top x)^2\right] = \frac{L_0^2}{2}h(w, w_*),
\end{aligned}
\tag{25}
$$

where the inequality is due to $L_0$-Lipschitzness of $\sigma(\cdot)$. We can now invoke Lemma 2 to conclude that the objective function is $\rho = \frac{2\alpha^2}{L_0^2}$-quasar convex.

$\square$

## C.2 PROOF OF LEMMA 4

**Lemma 4** *Assume that there exists a finite constant $C_R > 0$ such that all $w \in \mathbb{R}^d$ in the balls of radius $R$ centered at $\pm w_*$ satisfy $\mathbb{E}_{x \sim D}\left[\left((w + w_*)^\top x\right)^2 \|x\|_2^2\right] \le C_R$. Then, the loss function (11) is $\frac{1}{2}C_R$-generalized smooth w.r.t. $h(w, w_*) = \|w - w_*\|_2^2$.*

*Proof.* We have

$$
\begin{aligned}
f(w) - f(w_*) &= \mathbb{E}_{x \sim D}\left[\frac{1}{2}\left((w^\top x)^2 - (w_*^\top x)^2\right)^2\right] \\
&= \mathbb{E}_{x \sim D}\left[\frac{1}{2}\left((w^\top x) - (w_*^\top x)\right)^2\left((w^\top x) + (w_*^\top x)\right)^2\right] \\
&\le \frac{1}{2}\|w - w_*\|_2^2 \mathbb{E}_{x \sim D}\left[\left((w + w_*)^\top x\right)^2 \|x\|_2^2\right] \le \frac{1}{2}C_R\|w - w_*\|_2^2.
\end{aligned}
\tag{26}
$$

$\square$

## C.3 PROOF OF LEMMA 5

**Lemma 5** *When the link function is ReLU, the loss function (11) is $\frac{1}{2}\mathbb{E}_{x \sim D}[\|x\|_2^2]$-generalized smooth w.r.t. $h(w, w_*) = \|w - w_*\|_2^2$.*

*Proof.* We have

$$
\begin{aligned}
f(w) - f(w_*) &= \mathbb{E}_x\left[\frac{1}{2}\left(\sigma(w^\top x) - \sigma(w_*^\top x)\right)^2\right] \le \mathbb{E}_x\left[\frac{1}{2}\left(w^\top x - w_*^\top x\right)^2\right] \\
&\le \mathbb{E}_x\left[\frac{1}{2}\|w - w_*\|_2^2\|x\|_2^2\right] \\
&\le \frac{1}{2}\mathbb{E}_x[\|x\|_2^2]\|w - w_*\|_2^2,
\end{aligned}
\tag{27}
$$

where the first inequality uses that ReLU is 1-Lipschitz.

$\square$

## C.4 PROOF OF LEMMA 6

**Lemma 6** *Suppose that the function $f(\cdot)$ satisfies $C_v$-one-point convexity and $\hat{\rho}$-quasar convexity. Then, it is also $\left(\rho = \frac{\hat{\rho}}{\theta}, \mu = \frac{2C_v(\theta - 1)}{\hat{\rho}}\right)$-strongly quasar convex for any $\theta > 1$.*

*Proof.* We have

$$
\begin{aligned}
f(w) - f(w_*) &\le \frac{1}{\hat{\rho}}\langle \nabla f(w), w - w_*\rangle = \frac{\theta}{\hat{\rho}}\langle \nabla f(w), w - w_*\rangle - \frac{\theta - 1}{\hat{\rho}}\langle \nabla f(w), w - w_*\rangle \\
&\le \frac{\theta}{\hat{\rho}}\langle \nabla f(w), w - w_*\rangle - \frac{\theta - 1}{\hat{\rho}}C_v\|w - w_*\|^2,
\end{aligned}
\tag{28}
$$

where the last inequality uses the definition of $C_v$-one-point convexity. Rearranging the above inequality, we get $f(w_*) \ge f(w) + \frac{1}{\hat{\rho}/\theta}\langle \nabla f(w), w_* - w\rangle + \frac{2C_v(\theta - 1)/\hat{\rho}}{2}\|w_* - w\|^2$.

$\square$

## C.5 PROOF OF LEMMA 7 AND LEMMA 8

**Lemma 7** *Suppose that the function $f(\cdot)$ is $\nu$-QG and $\hat{\rho}$-quasar convex w.r.t. a global minimizer $w_*$. Then, it is also $(\rho = \hat{\rho}\theta, \mu = \frac{\nu(1 - \theta)}{\theta})$-strongly quasar convex for any $\theta < 1$.*

*Proof.* By $\hat{\rho}$-quasar convexity, we have

$$
\begin{aligned}
\langle \nabla f(w), w - w_* \rangle \geq \hat{\rho}(f(w) - f(w_*)) &= \hat{\rho}\theta(f(w) - f(w_*)) + \hat{\rho}(1 - \theta)(f(w) - f(w_*)) \\
&\geq \hat{\rho}\theta(f(w) - f(w_*)) + \frac{\hat{\rho}(1-\theta)\nu}{2}\|w - w_*\|^2,
\end{aligned}
\tag{29}
$$

where the last inequality uses the definition of $\nu$-QG. Rearranging the above inequality, we get $f(w_*) \geq f(w) + \frac{1}{\hat{\rho}\theta}\langle \nabla f(w), w_* - w \rangle + \frac{\nu(1-\theta)/\theta}{2}\|w_* - w\|^2$, which shows the result. $\qquad\square$

**Lemma 8** *Following the setting of Lemma 3, assume that the smallest eigenvalue of the matrix $\mathbb{E}_{x\sim\mathcal{D}}[xx^\top]$ satisfies $\lambda_{\min}(\mathbb{E}_{x\sim\mathcal{D}}[xx^\top]) > 0$. Then, the function (11) is $\alpha^2\lambda_{\min}(\mathbb{E}_{x\sim\mathcal{D}}[xx^\top])$-QG.*

*Proof.* We have

$$
\begin{aligned}
f(w) - f(w_*) &= \mathbb{E}_{x\sim\mathcal{D}}\left[\frac{1}{2}\left(\sigma(w^\top x) - \sigma(w_*^\top x)\right)^2\right] \\
&= \mathbb{E}_{x\sim\mathcal{D}}\left[\frac{1}{2}\left(\frac{\sigma(w^\top x) - \sigma(w_*^\top x)}{w^\top x - w_*^\top x}(w^\top x - w_*^\top x)\right)^2\right] \\
&\geq \frac{1}{2}\alpha^2\mathbb{E}_{x\sim\mathcal{D}}\left[(w^\top x - w_*^\top x)^2\right] \\
&= \frac{1}{2}\alpha^2(w - w_*)^\top\mathbb{E}_{x\sim\mathcal{D}}\left[xx^\top\right](w - w_*) \\
&\geq \frac{1}{2}\alpha^2\lambda_{\min}(\mathbb{E}_{x\sim D}[xx^\top])\|w - w_*\|^2,
\end{aligned}
\tag{30}
$$

where the second-to-last inequality uses that the derivative of the link function satisfies $\sigma'(\cdot) \geq \alpha$.

$\qquad\square$

## D  PROOF OF THEOREM 1 AND THEOREM 2

**Theorem 1** *Assume that the function $f(\cdot)$ is $L$-smooth and $\rho$-quasar convex. Let $\eta_t = \frac{2}{\rho t}, \eta_t' = 0, \gamma_t = \frac{1}{L}$, and $\gamma_t' = \frac{\rho t}{2L}$. Then, the update $w_t$ of the continuized algorithm (3) satisfies*

$$
\mathbb{E}[f(w_t) - f(w_*)] \leq \frac{2L\|z_0 - w_*\|^2}{\rho^2 t^2}.
$$

*Furthermore, for the update $\tilde{w}_k$ of the discrete-time algorithm (8)-(10), if the parameters are chosen as $\tau_k = 1 - \left(\frac{T_k}{T_{k+1}}\right)^{2/\rho}, \tau_k' = 0, \tilde{\gamma}_k = \frac{1}{L}$, and $\tilde{\gamma}_k' = \frac{\rho T_k}{2L}$, then*

$$
\mathbb{E}[T_k^2(f(\tilde{w}_k) - f(w_*))] \leq \frac{2L\|\tilde{z}_0 - w_*\|^2}{\rho^2}.
$$

**Theorem 2** *Assume that the function $f(\cdot)$ is $L$-smooth and $(\rho, \mu)$-strongly quasar convex, where $\mu > 0$. Let $\gamma_t = \frac{1}{L}, \gamma_t' = \frac{1}{\sqrt{\mu L}}, \eta_t' = \rho\sqrt{\frac{\mu}{L}}$, and $\eta_t = \sqrt{\frac{\mu}{L}}$. Then, the update $w_t$ of the continuized algorithm (3) satisfies*

$$
\mathbb{E}[f(w_t) - f(w_*)] \leq \left(f(w_0) - f(w_*) + \frac{\mu}{2}\|z_0 - w_*\|^2\right)\exp\left(-\rho\sqrt{\frac{\mu}{L}}t\right).
$$

*Furthermore, for the update $\tilde{w}_k$ of the discrete-time algorithm (8)-(10), if the parameters are chosen as $\tau_k = \frac{1}{1+\rho}\left(1 - \exp\left(-(1+\rho)\sqrt{\frac{\mu}{L}}(T_{k+1} - T_k)\right)\right), \tau_k' = \frac{\rho\left(1-\exp\left(-(1+\rho)\sqrt{\frac{\mu}{L}}(T_{k+1}-T_k)\right)\right)}{\rho+\exp\left(-(1+\rho)\sqrt{\frac{\mu}{L}}(T_{k+1}-T_k)\right)}, \tilde{\gamma}_k = \frac{1}{L}$, and $\tilde{\gamma}_k' = \frac{1}{\sqrt{\mu L}}$, then*

$$
\mathbb{E}[\exp\left(\rho\sqrt{\frac{\mu}{L}}T_k\right)(f(\tilde{w}_k) - f(w_*))] \leq f(\tilde{w}_0) - f(w_*) + \frac{\mu}{2}\|\tilde{z}_0 - w_*\|^2.
$$

The proof follows that of Theorem 2 in Even et al. (2021) with some modifications to account for (strong) quasar convexity. We will consider a Lyapunov function for the continuized process (3), defined as:

$$\phi_t := A_t \left( f(w_t) - f(w_*) \right) + \frac{B_t}{2} \|z_t - w_*\|^2. \tag{31}$$

We will show that $\phi_t$ is a super-martingale under certain choices of parameters $\eta_t, \eta_t', \gamma_t, \gamma_t', A_t$, and $B_t$. Let us first denote the process $\bar{w}_t := (t, w_t, z_t)$, whose dynamic is:

$$d\bar{w}_t = b(\bar{w}_t)dt + G(\bar{w}_t)dN(t), \quad b(\bar{w}_t) = \begin{bmatrix} 1 \\ \eta_t(z_t - w_t) \\ \eta_t'(w_t - z_t) \end{bmatrix}, \quad G(\bar{w}_t) = \begin{bmatrix} 0 \\ -\gamma_t \nabla f(w_t) \\ -\gamma_t' \nabla f(w_t) \end{bmatrix}. \tag{32}$$

Then, by Proposition 2 of Even et al. (2021), we have

$$\phi_t = \phi_0 + \int_0^t \langle \nabla \phi(\bar{w}_s), b(\bar{w}_s) \rangle ds + \int_0^t \left( \phi(\bar{w}_s + G(\bar{w}_t)) - \phi(\bar{w}_s) \right) ds + M_t, \tag{33}$$

where $M_t$ is a martingale. Therefore, to show $\phi_t$ is a supermartingale, it suffices to show:

$$I_t := \langle \nabla \phi(\bar{w}_t), b(\bar{w}_t) \rangle + \phi(\bar{w}_t + G(\bar{w}_t)) - \phi(\bar{w}_t) \leq 0. \tag{34}$$

For the first term of $I_t$. we have

$$\begin{aligned} \langle \nabla \phi(\bar{w}_t), b(\bar{w}_t) \rangle &= \partial_t \phi(\bar{w}_t) + \langle \partial_w \phi(\bar{w}_t), \eta_t(z_t - w_t) \rangle + \langle \partial_z \phi(\bar{w}_t), \eta_t'(w_t - z_t) \rangle \\ &= \frac{dA_t}{dt}(f(w_t) - f_*) + \frac{1}{2}\frac{dB_t}{dt}\|z_t - w_*\|^2 \\ &\quad + A_t \eta_t \langle \nabla f(w_t), z_t - w_t \rangle + B_t \eta_t' \langle z_t - w_*, z_t - w_t \rangle. \end{aligned} \tag{35}$$

By $(\rho, \mu)$-strongly quasar-convexity, we have

$$f(w_t) - f(w_*) \leq \frac{1}{\rho}\langle \nabla f(w_t), w_t - w_* \rangle - \frac{\mu}{2}\|w_t - w_*\|^2. \tag{36}$$

Furthermore, the following inequality holds,

$$\langle z_t - w_*, w_t - z_t \rangle \leq \frac{1}{2}(\|w_t - w_*\|^2 - \|z_t - w_*\|^2), \tag{37}$$

because $\langle z_t - w_*, w_t - z_t \rangle = \langle z_t - w_*, w_t - w_* \rangle - \|z_t - w_*\|^2 \leq \|z_t - w_*\|\|w_t - w_*\| - \|z_t - w_*\|^2 \leq \frac{1}{2}(\|w_t - w_*\|^2 - \|z_t - w_*\|^2)$. Combining (35)-(37) gives

$$\begin{aligned} \langle \nabla \phi(\bar{w}_t), b(\bar{w}_t) \rangle \leq{} & \left( \frac{1}{\rho}\frac{dA_t}{dt} - A_t \eta_t \right) \langle \nabla f(w_t), w_t - w_* \rangle + \left( B_t \eta_t' - \frac{dA_t}{dt}\mu \right) \frac{1}{2}\|w_t - w_*\|^2 \\ & + \left( \frac{dB_t}{dt} - B_t \eta_t' \right) \frac{1}{2}\|z_t - w_*\|^2 + A_t \eta_t \langle \nabla f(w_t), z_t - w_* \rangle. \end{aligned} \tag{38}$$

For the second term of $I_t$, we have

$$\begin{aligned} \phi(\bar{w}_t + G(\bar{w}_t)) - \phi(\bar{w}_t) ={} & A_t \left( f(w_t - \gamma_t \nabla f(w_t)) - f_* \right) \\ & + \frac{B_t}{2} \left( \|z_t - \gamma_t' \nabla f(w_t) - w_* + \gamma_t \nabla f(w_t)\|^2 - \|z_t - w_*\|^2 \right). \end{aligned} \tag{39}$$

Since by smoothness, we have

$$f(w_t - \gamma_t \nabla f(w_t)) - f(w_t) \leq \langle \nabla f(w_t), -\gamma_t \nabla f(w_t) \rangle + \frac{L}{2}\|\gamma_t \nabla f(w_t)\|^2 = -\gamma_t(2 - L\gamma_t)\frac{1}{2}\|\nabla f(w_t)\|^2. \tag{40}$$

So the second term can be bounded as

$$\phi(\bar{w}_t + G(\bar{w}_t)) - \phi(\bar{w}_t) \leq \left( B_t(\gamma_t')^2 - A_t \gamma_t(2 - L\gamma_t) \right) \frac{1}{2}\|\nabla f(w_t)\|^2 - \beta\gamma_t' \langle z_t - w_*, \nabla f(w_t) \rangle. \tag{41}$$

Combining (34), (38), (41), we have

$$
\begin{aligned}
I_t \leq {}& \left( \frac{1}{\rho} \frac{dA_t}{dt} - A_t \eta_t \right) \langle \nabla f(w_t), w_t - w_* \rangle + \left( B_t \eta_t' - \frac{dA_t}{dt} \mu \right) \frac{1}{2} \| w_t - w_* \|^2 \\
& + \left( \frac{dB_t}{dt} - B_t \eta_t' \right) \frac{1}{2} \| z_t - w_* \|^2 + (A_t \eta_t - B_t \gamma_t') \langle \nabla f(w_t), z_t - w_* \rangle \\
& + \left( B_t (\gamma_t')^2 - A_t \gamma_t (2 - L\gamma_t) \right) \frac{1}{2} \| \nabla f(w_t) \|^2 .
\end{aligned}
\tag{42}
$$

Now let us determine the parameters $\eta_t$, $\eta_t'$, $\gamma_t$, $\gamma_t'$, $A_t$ and $B_t$. We start by taking $\gamma_t = \frac{1}{L}$. Since we need $I_t \leq 0$, we want to satisfy

$$
\frac{1}{\rho} \frac{dA_t}{dt} = A_t \eta_t, \quad \frac{dB_t}{dt} = B_t \eta_t', \quad A_t \eta_t = B_t \gamma_t', \quad B_t \eta_t' = \frac{dA_t}{dt} \mu, \quad B_t (\gamma_t')^2 = \frac{A_t}{L}.
\tag{43}
$$

Let us choose

$$
\gamma_t' = \sqrt{\frac{A_t}{LB_t}}, \quad \eta_t = \frac{B_t \gamma_t'}{A_t} = \sqrt{\frac{B_t}{LA_t}}, \quad \eta_t' = \frac{dA_t}{dt} \frac{\mu}{B_t} = \frac{\rho A_t \eta_t \mu}{B_t} = \rho \mu \sqrt{\frac{A_t}{LB_t}},
\tag{44}
$$

which ensures that the last three conditions of (43) are satisfied. It remains to show that the first two hold:

$$
\frac{1}{\rho} \frac{dA_t}{dt} = A_t \eta_t, \quad \frac{dB_t}{dt} = B_t \eta_t' = \rho \mu \sqrt{\frac{A_t B_t}{L}}.
\tag{45}
$$

We have

$$
\frac{d}{dt}(\sqrt{A_t}) = \frac{1}{2\sqrt{A_t}} \frac{dA_t}{dt} \overset{(a)}{=} \frac{\rho}{2} \sqrt{\frac{B_t}{L}}, \quad \frac{d}{dt}(\sqrt{B_t}) = \frac{1}{2\sqrt{B_t}} \frac{dB_t}{dt} \overset{(b)}{=} \frac{\rho \mu}{2} \sqrt{\frac{A_t}{L}},
\tag{46}
$$

where (a) uses that $\frac{dA_t}{dt} = \rho A_t \eta_t = \rho A_t \sqrt{\frac{B_t}{LA_t}}$ from (45), and (b) uses $\frac{dB_t}{dt} = \rho \mu \sqrt{\frac{A_t B_t}{L}}$ from (45).

The equations on (46) imply that

$$
\frac{d^2}{dt^2}(\sqrt{A_t}) = \frac{\rho^2 \mu}{4L} \sqrt{A_t}, \quad \sqrt{B_t} = \frac{2\sqrt{L}}{\rho} \frac{d}{dt}(\sqrt{A_t}).
\tag{47}
$$

### D.1 $\rho$-QUASAR-CONVEX

*Proof.* (of Theorem 1)

For the case of $\mu = 0$, we choose $A_0 = 0$ and $B_0 = 1$. From (46), we have $\frac{d}{dt}(\sqrt{B_t}) = 0$ so that $B_t = 1$ and that $\frac{d}{dt}(\sqrt{A_t}) = \frac{\rho}{2\sqrt{L}}$, consequently, $\sqrt{A_t} = \rho \frac{t}{2\sqrt{L}}$. From (44), we conclude $\gamma_t = \frac{1}{L}$, $\gamma_t' = \frac{\rho t}{2L}$, $\eta_t' = 0$, and $\eta_t = \frac{2}{\rho t}$. Therefore, as $\phi_t$ is a super-martingale, we get

$$
\mathbb{E}[A_T(f(w_T) - f_*)] \leq \mathbb{E}[\phi_T] \leq \phi_0 = \| z_0 - w_* \|^2
\tag{48}
$$

So we have

$$
\mathbb{E}[f(w_T) - f_*] \leq \frac{4L \| z_0 - w_* \|^2}{\rho^2 t^2}.
\tag{49}
$$

This proves the first part of Theorem 1.

The ODEs (4)-(5) become

$$
dw_t = \eta_t (z_t - w_t) dt = \frac{2}{\rho t}(z_t - w_t) dt
\tag{50}
$$

$$
dz_t = \eta_t'(w_t - z_t) dt = 0.
\tag{51}
$$

Integrating the ODEs from time $t_0$ to $t$,

$$
w_t = z_{t_0} + \left( \frac{t_0}{t} \right)^{2/\rho} (w_{t_0} - z_{t_0})
\tag{52}
$$

$$
z_t = z_{t_0}.
\tag{53}
$$

Using Lemma 1 with $t_0 = T_k$, $t = T_{k+1-}$, (52) becomes

$$\tilde{v}_k = \tilde{z}_k \left( 1 - \left( \frac{T_k}{T_{k+1}} \right)^{2/\rho} \right) + \tilde{w}_k \left( \frac{T_k}{T_{k+1}} \right)^{2/\rho} \tag{54}$$

This together with (8) implies that $\tau_k = 1 - \left( \frac{T_k}{T_{k+1}} \right)^{2/\rho}$, while comparing (20) and (53) leads to $\tau_k'' = 0$ and $\tau_k' = \frac{\tau_k''}{1-\tau_k} = 0$. Moreover, we have $\tilde{\gamma}_k = \gamma_{T_k} = \frac{1}{L}$ and $\tilde{\gamma}_k' = \gamma_{T_k}' = \frac{\rho T_k}{2L}$. This proves the second part of Theorem 1.

$\square$

### D.2  $(\rho, \mu)$-STRONGLY QUASAR-CONVEX

*Proof.* (of Theorem 2)

From (44), we choose $\gamma_t = \frac{1}{L}$, $\gamma_t' = \frac{1}{\sqrt{\mu L}}$, $\eta_t' = \rho \sqrt{\frac{\mu}{L}}$, $\eta_t = \sqrt{\frac{\mu}{L}}$. We have

$$\sqrt{A_t} = \sqrt{A_0} \exp \left( \frac{\rho}{2} \sqrt{\frac{\mu}{L}} t \right), \qquad \sqrt{B_t} = \sqrt{A_0} \sqrt{\mu} \exp \left( \frac{\rho}{2} \sqrt{\frac{\mu}{L}} t \right). \tag{55}$$

Then, we can conclude that

$$\mathbb{E}[A_T(f(w_T) - f_*)] \leq \mathbb{E}[\phi_T] \leq \phi_0 = A_0 \left( f(w_0) - f(w_*) \right) + A_0 \frac{\mu}{2} \| z_0 - w_* \|^2 \tag{56}$$

So we have

$$\mathbb{E}[f(w_T) - f_*] \leq \left( (f(w_0) - f(w_*)) + \frac{\mu}{2} \| z_0 - w_* \|^2 \right) \exp \left( -\rho \sqrt{\frac{\mu}{L}} t \right). \tag{57}$$

This proves the first part of Theorem 2.

The ODEs (4)-(5) become

$$dw_t = \eta_t(z_t - w_t)dt = \sqrt{\frac{\mu}{L}}(z_t - w_t)dt \tag{58}$$

$$dz_t = \eta_t'(w_t - z_t)dt = \rho \sqrt{\frac{\mu}{L}}(w_t - z_t)dt. \tag{59}$$

The solutions are

$$w_t = \frac{\rho w_{t_0} + z_{t_0}}{1 + \rho} + \frac{w_{t_0} - z_{t_0}}{1 + \rho} \exp \left( -(1 + \rho) \sqrt{\frac{\mu}{L}}(t - t_0) \right) \tag{60}$$

$$= w_{t_0} + \frac{1}{1 + \rho} \left( 1 - \exp \left( -(1 + \rho) \sqrt{\frac{\mu}{L}}(t - t_0) \right) \right) (z_{t_0} - w_{t_0}) \tag{61}$$

$$z_t = \frac{\rho w_{t_0} + z_{t_0}}{1 + \rho} + \rho \frac{z_{t_0} - w_{t_0}}{1 + \rho} \exp \left( -(1 + \rho) \sqrt{\frac{\mu}{L}}(t - t_0) \right) \tag{62}$$

$$= z_{t_0} + \frac{\rho}{1 + \rho} \left( 1 - \exp \left( -(1 + \rho) \sqrt{\frac{\mu}{L}}(t - t_0) \right) \right) (w_{t_0} - z_{t_0}). \tag{63}$$

Taking $t_0 = T_k$, $t = T_{k+1-}$, the above becomes

$$\tilde{v}_k = \tilde{w}_k + \frac{1}{1 + \rho} \left( 1 - \exp \left( -(1 + \rho) \sqrt{\frac{\mu}{L}}(T_{k+1} - T_k) \right) \right) (\tilde{z}_k - \tilde{w}_k) \tag{64}$$

$$\tilde{z}_{T_{k+1-}} = \tilde{z}_k + \frac{\rho}{1 + \rho} \left( 1 - \exp \left( -(1 + \rho) \sqrt{\frac{\mu}{L}}(T_{k+1} - T_k) \right) \right) (\tilde{w}_k - \tilde{z}_k) \tag{65}$$

Comparing equation (64) and (14), we know that $\tau_k = \frac{1}{1+\rho} \left( 1 - \exp \left( -(1+\rho) \sqrt{\frac{\mu}{L}}(T_{k+1} - T_k) \right) \right)$. Furthermore, by using (64), we get

$$\tilde{w}_k - \tilde{z}_k = \frac{1}{1 - \frac{1}{1+\rho} \left( 1 - \exp \left( -(1+\rho) \sqrt{\frac{\mu}{L}}(T_{k+1} - T_k) \right) \right)} (\tilde{v}_k - \tilde{z}_k). \tag{66}$$

Based on (65), (66), and (21), we conclude that $\tau_k' = \frac{\rho\left(1-\exp\left(-(1+\rho)\sqrt{\frac{\mu}{L}}(T_{k+1}-T_k)\right)\right)}{\rho+\exp\left(-(1+\rho)\sqrt{\frac{\mu}{L}}(T_{k+1}-T_k)\right)}$. Moreover, we have $\tilde{\gamma}_k = \gamma_{T_k} = \frac{1}{L}$ and $\tilde{\gamma}_k' = \gamma_{T_k}' = \frac{1}{\sqrt{\mu L}}$. This proves the second part of Theorem 2.

$\square$

### D.3 PROOF OF COROLLARY 1 AND COROLLARY 2

**Corollary 1:** *The update $\tilde{w}_k$ of the algorithm (8)-(10) with the same parameters indicated in Theorem 1 satisfies $f(\tilde{w}_k) - f(w_*) \leq \frac{2c_0 L\|\tilde{z}_0 - w_*\|^2}{(1-c)^2 \rho^2 k^2}$, with probability at least $1 - \frac{1}{c^2 k} - \frac{1}{c_0}$ for any $c \in (0,1)$ and $c_0 > 1$.*

*Proof.* Using Markov's inequality and Theorem 1, we get

$$\Pr\left[T_k^2\left(f(\tilde{w}_k) - f(w_*)\right) \geq C_0\right] \leq \frac{\mathbb{E}\left[T_k^2(f(\tilde{w}_k)-f(w_*))\right]}{C_0} \leq \frac{2L\|\tilde{z}_0-w_*\|^2/\rho^2}{C_0}. \tag{67}$$

Let $C_0 := c_0 2L\|\tilde{z}_0 - w_*\|^2/\rho^2$, where $c_0 > 1$ is a universal constant. Then, with probability $1 - \frac{1}{c_0}$,

$$T_k^2\left(f(\tilde{w}_k) - f(w_*)\right) \leq \frac{2c_0 L\|\tilde{z}_0-w_*\|^2}{\rho^2}. \tag{68}$$

By Chebyshev's inequality, we have $\Pr\left(|T_k - \mathbb{E}[T_k]| \geq c\mathbb{E}[T_k]\right) \leq \frac{\text{Var}(T_k)}{c^2(\mathbb{E}[T_k])^2}$, where $c > 0$ is a universal constant. Hence, we have $T_k \geq (1-c)\mathbb{E}[T_k] = (1-c)k$ with probability at least $1 - \frac{1}{c^2 k}$, where we used the fact that $\mathbb{E}[T_k] = \text{Var}[T_k] = k$ as $T_k$ is the sum of $k$ Poisson random variables with mean 1. Combining this lower bound of $T_k$ and (68) leads to the result. $\square$

**Corollary 2:** *The update $\tilde{w}_k$ of the algorithm (8)-(10) with the same parameters indicated in Theorem 2 satisfies $f(\tilde{w}_k) - f(w_*) \leq c_0\left(f(\tilde{w}_0) - f(w_*) + \frac{\mu}{2}\|\tilde{z}_0 - w_*\|^2\right)\exp\left(-\rho\sqrt{\frac{\mu}{L}}(1-c)k\right)$, with probability at least $1 - \frac{1}{c^2 k} - \frac{1}{c_0}$ for any $c \in (0,1)$ and $c_0 > 1$.*

*Proof.* Using Markov's inequality and Theorem 2, we get

$$\Pr\left[\exp\left(\rho\sqrt{\frac{\mu}{L}}T_k\right)\left(f(\tilde{w}_k) - f(w_*)\right) \geq C_0\right] \leq \frac{\mathbb{E}\left[\exp\left(\rho\sqrt{\frac{\mu}{L}}T_k\right)(f(\tilde{w}_k)-f(w_*))\right]}{C_0} \leq \frac{f(\tilde{w}_0)-f(w_*)+\frac{\mu}{2}\|\tilde{z}_0-w_*\|^2}{C_0}. \tag{69}$$

Let $C_0 := c_0\left(f(\tilde{w}_0) - f(w_*) + \frac{\mu}{2}\|\tilde{z}_0 - w_*\|^2\right)$, where $c_0 > 1$ is a universal constant. Then, with probability $1 - \frac{1}{c_0}$,

$$\exp\left(\rho\sqrt{\frac{\mu}{L}}T_k\right)\left(f(\tilde{w}_k) - f(w_*)\right) \leq c_0\left((f(\tilde{w}_0) - f(w_*)) + \frac{\mu}{2}\|\tilde{z}_0 - w_*\|^2\right) \tag{70}$$

By Chebyshev's inequality, we have $\Pr\left(|T_k - \mathbb{E}[T_k]| \geq c\mathbb{E}[T_k]\right) \leq \frac{\text{Var}(T_k)}{c^2(\mathbb{E}[T_k])^2}$, where $c > 0$ is a universal constant. Hence, we have $T_k \geq (1-c)\mathbb{E}[T_k] = (1-c)k$ with probability at least $1 - \frac{1}{c^2 k}$, where we used the fact that $\mathbb{E}[T_k] = \text{Var}[T_k] = k$ as $T_k$ is the sum of $k$ Poisson random variables with mean 1. Combining this lower bound of $T_k$ and (70) leads to the result. $\square$

## E  PROOF OF THEOREM 3

**Theorem 3 (Continuized algorithm (13) for GLMs)** *Choose $\eta_t' = \sqrt{\frac{\mu}{\tilde{\kappa}R^2}}$, $\eta_t = \sqrt{\frac{\mu}{\tilde{\kappa}R^2}}$, $\gamma_t = \frac{1}{R^2}$, and $\gamma_t' = \frac{1}{\sqrt{\mu\tilde{\kappa}R^2}}$. Then, the update $w_t$ of (13) satisfies*

$$\mathbb{E}\left[\frac{1}{2}\|w_t - w_*\|^2\right] \leq \frac{1}{2}\left(\|w_0 - w_*\|^2 + \mu\|z_0 - w_*\|^2_{H(w_0)^{-1}}\right)\exp\left(-\sqrt{\frac{\mu}{\tilde{\kappa}R^2}}t\right).$$

*Proof.* Let us denote $H_t := H(w_t) = \mathbb{E}_x[\psi(w_t^\top x, w_*^\top x)xx^\top]$ and consider a Lyapunov function for the continuized process (3), defined as:

$$\phi_t := \frac{A_t}{2}\|w_t - w_*\|^2 + \frac{B_t}{2}\|z_t - w_*\|^2_{H_t^{-1}}. \tag{71}$$

We first show that $\phi_t$ is a super-martingale under certain values of parameters $\eta_t$, $\eta_t'$, $\gamma_t$, $\gamma_t'$, $A_t$, and $B_t$. Let us denote the process $\bar{w}_t := (t, w_t, z_t)$, which satisfies the following equation:

$$d\bar{w}_t = b(\bar{w}_t)dt + \int_\Xi G(\bar{w}_t; \xi)dN(t, \xi), \quad b(\bar{w}_t) = \begin{bmatrix} 1 \\ \eta_t(z_t - w_t) \\ \eta_t'(w_t - z_t) \end{bmatrix}, \quad G(\bar{w}_t; \xi) = \begin{bmatrix} 0 \\ -\gamma_t g(w_t; \xi) \\ -\gamma_t' g(w_t; \xi) \end{bmatrix}. \tag{72}$$

Then, by Proposition 2 of Even et al. (2021), we have

$$\phi_t = \phi_0 + \int_0^t I_s ds + M_t, \tag{73}$$

where $M_t$ is a martingale and $I_t$ is

$$I_t := \langle \nabla \phi(\bar{w}_t), b(\bar{w}_t) \rangle + \mathbb{E}_\xi[\phi(\bar{w}_t + G(\bar{w}_t; \xi)) - \phi(\bar{w}_t)] \tag{74}$$

For the first term of $I_t$, we have

$$\langle \nabla \phi(\bar{w}_t), b(\bar{w}_t) \rangle = \partial_t \phi(\bar{w}_t) + \langle \partial_w \phi(\bar{w}_t), \eta_t(z_t - w_t) \rangle + \langle \partial_z \phi(\bar{w}_t), \eta_t'(w_t - z_t) \rangle$$

$$= \frac{1}{2}\frac{dA_t}{dt}\|w_t - w_*\|^2 + \frac{1}{2}\frac{dB_t}{dt}\|z_t - w_*\|^2_{H_t^{-1}} + A_t\eta_t\langle w_t - w_*, z_t - w_t \rangle + B_t\eta_t'\langle z_t - w_*, H_t^{-1}(w_t - z_t) \rangle. \tag{75}$$

Since $H_t = \mathbb{E}_x[\psi(w_t^\top x, w_*^\top x)xx^\top] \succeq \mu I_d$, we have

$$\frac{\mu}{2}\|w_t - w_*\|^2_{H_t^{-1}} \le \frac{1}{2}\|w_t - w_*\|^2. \tag{76}$$

Using (76), we have

$$\frac{1}{2}\|w_t - w_*\|^2 \le \|w_t - w_*\|^2 - \frac{\mu}{2}\|w_t - w_*\|^2_{H_t^{-1}}. \tag{77}$$

Furthermore, the following inequalities hold,

$$\langle z_t - w_*, H_t^{-1}(w_t - z_t) \rangle \le \frac{1}{2}(\|w_t - w_*\|^2_{H_t^{-1}} - \|z_t - w_*\|^2_{H_t^{-1}}). \tag{78}$$

This is because $\langle z_t - w_*, H_t^{-1}(w_t - z_t) \rangle = \langle z_t - w_*, H_t^{-1}(w_t - w_*) \rangle - \|z_t - w_*\|^2_{H_t^{-1}} \le \frac{1}{2}\left(\|w_t - w_*\|^2_{H_t^{-1}} - \|z_t - w_*\|^2_{H_t^{-1}}\right)$.

Combining (75)-(78), we get

$$\langle \nabla \phi(\bar{w}_t), b(\bar{w}_t) \rangle \le \frac{dA_t}{dt}\|w_t - w_*\|^2 + \left(B_t\eta_t' - \frac{dA_t}{dt}\mu\right)\frac{1}{2}\|w_t - w_*\|^2_{H_t^{-1}}$$

$$+ \left(\frac{dB_t}{dt} - B_t\eta_t'\right)\frac{1}{2}\|z_t - w_*\|^2_{H_t^{-1}} + A_t\eta_t\langle w_t - w_*, z_t - w_t \rangle. \tag{79}$$

For the second term of $I_t$, we have

$$\mathbb{E}_\xi[\phi(\bar{w}_t + G(\bar{w}_t; \xi)) - \phi(\bar{w}_t)]$$

$$= \mathbb{E}_\xi\left[\frac{A_t}{2}\left(\|w_t - \gamma_t g(w_t; \xi) - w_*\|^2 - \|w_t - w_*\|^2\right) + \frac{B_t}{2}\left(\|z_t - \gamma_t' g(w_t; \xi) - w_*\|^2_{H_t^{-1}} - \|z_t - w_*\|^2_{H_t^{-1}}\right)\right]$$

$$= \mathbb{E}_\xi\left[\frac{A_t\gamma_t^2}{2}\|g(w_t; \xi)\|^2 - A_t\gamma_t\langle w_t - w_*, g(w_t; \xi) \rangle + \frac{B_t(\gamma_t')^2}{2}\|g(w_t; \xi)\|^2_{H_t^{-1}} - B_t\gamma_t'\langle z_t - w_*, H_t^{-1}g(w_t; \xi) \rangle\right]. \tag{80}$$

Let us upper-bound the first two terms in (80). We have

$$\mathbb{E}_\xi[\|g(w_t; \xi)\|^2] = \mathbb{E}_x[(\sigma(w_t^\top x) - y)x, (\sigma(w_t^\top x) - y)x]$$

$$= \mathbb{E}_x[\psi(w_t^\top x, w_*^\top x)^2((w_t - w_*)^\top x)^2\|x\|^2]$$

$$= \langle w_t - w_*, \mathbb{E}_x[\psi(w_t^\top x, w_*^\top x)^2\|x\|^2xx^\top]w_t - w_* \rangle$$

$$\le R^2\langle w_t - w_*, \mathbb{E}_x[\psi(w_t^\top x, w_*^\top x)xx^\top]w_t - w_* \rangle = R^2\|w_t - w_*\|^2_{H_t}, \tag{81}$$

where in the last inequality we used

$$\mathbb{E}_x \left[ \psi(w_t^\top x, w_*^\top x)^2 \|x\|^2 xx^\top \right] \preceq R^2 \mathbb{E}_x \left[ \psi(w_t^\top x, w_*^\top x) xx^\top \right] = R^2 H_t. \tag{82}$$

Furthermore, we have

$$\begin{aligned}
\mathbb{E}_\xi[\langle g(w_t; \xi), w_t - w_* \rangle] &= \mathbb{E}_x[\langle \left( \sigma(w^\top x) - \sigma(w_*^\top x) \right) x, w_t - w_* \rangle] \\
&= (w_t - w_*)^\top \mathbb{E}_x[\psi(w^\top x, w_*^\top x) xx^\top](w_t - w_*) = \|w_t - w_*\|_{H_t}^2.
\end{aligned} \tag{83}$$

Therefore, the first two terms in (80) can be bounded as

$$\mathbb{E}_\xi \left[ \frac{A_t \gamma_t^2}{2} \|g(w_t; \xi)\|^2 - A_t \gamma_t \langle w_t - w_*, g(w_t; \xi) \rangle \right] \leq \left( \frac{A_t \gamma_t^2 R^2}{2} - A_t \gamma_t \right) \|w_t - w_*\|_{H_t}^2. \tag{84}$$

Now let us switch to upper-bound the last two terms in (80). We have

$$\begin{aligned}
\mathbb{E}_\xi[\|g(w_t; \xi)\|_{H_t^{-1}}^2] &= \mathbb{E}_x \left[ \left( \sigma(w_t^\top x) - y \right) x, H_t^{-1} \left( \sigma(w_t^\top x) - y \right) x \right] \\
&= \mathbb{E}_x \left[ \psi(w_t^\top x, w_*^\top x)^2 ((w_t - w_*)^\top x)^2 \|x\|_{H_t^{-1}}^2 \right] \\
&= \langle w_t - w_*, \mathbb{E}_x \left[ \psi(w_t^\top x, w_*^\top x)^2 \|x\|_{H_t^{-1}}^2 xx^\top \right], w_t - w_* \rangle \\
&\leq \tilde{\kappa} \mathbb{E}_x[\langle w_t - w_*, \psi(w_t^\top x, w_*^\top x) xx^\top (w_t - w_*) \rangle] = \tilde{\kappa} \|w_t - w_*\|_{H_t}^2,
\end{aligned} \tag{85}$$

where in the last inequality, we used the assumption that

$$\mathbb{E}_x \left[ \psi(w_t^\top x, w_*^\top x)^2 \|x\|_{H_t^{-1}}^2 xx^\top \right] \preceq \tilde{\kappa} \mathbb{E}_x[\psi(w_t^\top x, w_*^\top x) xx^\top] = \tilde{\kappa} H_t. \tag{86}$$

We also have

$$\begin{aligned}
\mathbb{E}_\xi \left[ \gamma_t' \langle z_t - w_*, H_t^{-1} g(w_t, \xi) \rangle \right] &= \mathbb{E}_\xi \left[ \gamma_t' \langle z_t - w_*, H_t^{-1} \left( \sigma(w_t^\top x) - \sigma(w_*^\top x) \right) x \rangle \right] \\
&= \mathbb{E}_\xi \left[ \gamma_t' \langle z_t - w_*, H_t^{-1} \psi(w_t^\top x, w_*^\top x) xx^\top (w_t - w_*) \rangle \right] \\
&= \gamma_t' \langle z_t - w_*, w_t - w_* \rangle.
\end{aligned} \tag{87}$$

So the last two terms in (80) can be bounded as

$$\mathbb{E}_\xi \left[ \frac{B_t (\gamma_t')^2}{2} \|g(w_t; \xi)\|_{H_t^{-1}}^2 - B_t \gamma_t' \langle z_t - w_*, H_t^{-1} g(w_t; \xi) \rangle \right] \leq \frac{B_t (\gamma_t')^2 \tilde{\kappa}}{2} \|w_t - w_*\|_{H_t}^2 - B_t \gamma_t' \langle z_t - w_*, w_t - w_* \rangle. \tag{88}$$

Therefore, combining (84), (88), and (80), we have

$$\begin{aligned}
&\mathbb{E}_\xi[\phi(\bar{w}_t + G(\bar{w}_t; \xi)) - \phi(\bar{w}_t)] \\
&\leq \left( \frac{A_t \gamma_t^2 R^2}{2} - A_t \gamma_t + \frac{B_t (\gamma_t')^2 \tilde{\kappa}}{2} \right) \|w_t - w_*\|_{H_t}^2 - B_t \gamma_t' \langle z_t - w_*, w_t - w_* \rangle.
\end{aligned} \tag{89}$$

Combining (74), (79), and (89), we have:

$$\begin{aligned}
I_t &\leq \left( \frac{dA_t}{dt} - A_t \eta_t \right) \|w_t - w_*\|^2 + \left( B_t \eta_t' - \frac{dA_t}{dt} \mu \right) \frac{1}{2} \|w_t - w_*\|_{H_t^{-1}}^2 \\
&\quad + \left( \frac{dB_t}{dt} - B_t \eta_t' \right) \frac{1}{2} \|z_t - w_*\|_{H_t^{-1}}^2 + (A_t \eta_t - B_t \gamma_t') \langle w_t - w_*, z_t - w_* \rangle \\
&\quad + \left( \frac{A_t \gamma_t^2 R^2}{2} - A_t \gamma_t + \frac{B_t (\gamma_t')^2 \tilde{\kappa}}{2} \right) \|w_t - w_*\|_{H_t}^2.
\end{aligned} \tag{90}$$

Now let us determine $\eta_t, \eta_t', \gamma_t, \gamma_t', A_t,$ and $B_t$. We start by taking $\gamma_t = \frac{1}{R^2}$. We want $I_t \leq 0$, so we want to satisfy

$$\frac{dA_t}{dt} = A_t \eta_t, \quad \frac{dB_t}{dt} = B_t \eta_t', \quad A_t \eta_t = B_t \gamma_t', \quad B_t \eta_t' = \frac{dA_t}{dt} \mu, \quad B_t (\gamma_t')^2 = \frac{A_t}{\tilde{\kappa} R^2}. \tag{91}$$

Let us choose

$$\gamma_t' = \sqrt{\frac{A_t}{B_t \tilde{\kappa} R^2}}, \quad \eta_t = \frac{B_t \gamma_t'}{A_t} = \sqrt{\frac{B_t}{\tilde{\kappa} R^2 A_t}}, \quad \eta_t' = \frac{dA_t}{dt} \frac{\mu}{B_t} = \frac{A_t \eta_t \mu}{B_t} = \mu \sqrt{\frac{A_t}{\tilde{\kappa} R^2 B_t}}, \quad (92)$$

which ensures that the last three conditions of (91) are satisfied. It remains to show that the first two hold:

$$\frac{dA_t}{dt} = A_t \eta_t, \quad \frac{dB_t}{dt} = B_t \eta_t' = \mu \sqrt{\frac{A_t B_t}{\tilde{\kappa} R^2}}. \quad (93)$$

We have

$$\frac{d}{dt}(\sqrt{A_t}) = \frac{1}{2\sqrt{A_t}} \frac{dA_t}{dt} \overset{(a)}{=} \frac{1}{2} \sqrt{\frac{B_t}{\tilde{\kappa} R^2}}, \quad \frac{d}{dt}(\sqrt{B_t}) = \frac{1}{2\sqrt{B_t}} \frac{dB_t}{dt} \overset{(b)}{=} \frac{\mu}{2} \sqrt{\frac{A_t}{\tilde{\kappa} R^2}}, \quad (94)$$

where (a) uses that $\frac{dA_t}{dt} = A_t \eta_t = A_t \sqrt{\frac{B_t}{\tilde{\kappa} R^2 A_t}}$ and (b) uses that $\frac{dB_t}{dt} = B_t \eta_t' = \mu \sqrt{\frac{A_t}{\tilde{\kappa} R^2 B_t}}$ from (92) and (93). The equations on (94) imply that

$$\frac{d^2}{dt^2}(\sqrt{A_t}) = \frac{\mu}{4\tilde{\kappa} R^2} \sqrt{A_t}, \quad \sqrt{B_t} = 2\sqrt{\tilde{\kappa} R^2} \frac{d}{dt}(\sqrt{A_t}). \quad (95)$$

Let us choose $\gamma_t = \frac{1}{R^2}, \gamma_t' = \frac{1}{\sqrt{\mu \tilde{\kappa} R^2}}, \eta_t' = \sqrt{\frac{\mu}{\tilde{\kappa} R^2}}, \eta_t = \sqrt{\frac{\mu}{\tilde{\kappa} R^2}}$. We have

$$\sqrt{A_t} = \sqrt{A_0} \exp\left(\frac{1}{2}\sqrt{\frac{\mu}{\tilde{\kappa} R^2}} t\right), \quad \sqrt{B_t} = \sqrt{A_0}\sqrt{\mu} \exp\left(\frac{1}{2}\sqrt{\frac{\mu}{\tilde{\kappa} R^2}} t\right). \quad (96)$$

Then, we can conclude that

$$\mathbb{E}[A_t \frac{1}{2}\|w_t - w_*\|^2] \leq \mathbb{E}[\phi_t] \leq \phi_0 = \left(\frac{A_0}{2}\|w_0 - w_*\|^2 + \frac{B_0}{2}\|z_0 - w_*\|_{H(w_0)^{-1}}^2\right). \quad (97)$$

So we have

$$\mathbb{E}[\frac{1}{2}\|w_t - w_*\|^2] \leq \frac{1}{A_0}\left(\frac{A_0}{2}\|w_0 - w_*\|^2 + \frac{B_0}{2}\|z_0 - w_*\|_{H(w_0)^{-1}}^2\right) \exp\left(-\sqrt{\frac{\mu}{\tilde{\kappa} R^2}} t\right), \quad (98)$$

and we have $\frac{B_0}{A_0} = \mu$ from (96) and we can choose $A_0 = 1$. This proves Theorem 3.

Now let us switch to determine the corresponding parameters of the discrete-time algorithm (8)-(10), where the gradient $\nabla(w_t)$ is now replaced with the stochastic pseudo-gradient $g(w_t; \xi_t)$. The ODEs (4)-(5) become

$$dw_t = \eta_t(z_t - w_t)dt = \sqrt{\frac{\mu}{\tilde{\kappa} R^2}}(z_t - w_t)dt \quad (99)$$

$$dz_t = \eta_t'(w_t - z_t)dt = \sqrt{\frac{\mu}{\tilde{\kappa} R^2}}(w_t - z_t)dt. \quad (100)$$

The solutions are

$$w_t = \frac{w_{t_0} + z_{t_0}}{2} + \frac{w_{t_0} - z_{t_0}}{2} \exp\left(-2\sqrt{\frac{\mu}{\tilde{\kappa} R^2}}(t - t_0)\right) \quad (101)$$

$$= w_{t_0} + \frac{1}{2}\left(1 - \exp\left(-2\sqrt{\frac{\mu}{\tilde{\kappa} R^2}}(t - t_0)\right)\right)(z_{t_0} - w_{t_0}) \quad (102)$$

$$z_t = \frac{w_{t_0} + z_{t_0}}{2} + \frac{z_{t_0} - w_{t_0}}{2} \exp\left(-2\sqrt{\frac{\mu}{\tilde{\kappa} R^2}}(t - t_0)\right) \quad (103)$$

$$= z_{t_0} + \frac{1}{2}\left(1 - \exp\left(-2\sqrt{\frac{\mu}{\tilde{\kappa} R^2}}(t - t_0)\right)\right)(w_{t_0} - z_{t_0}). \quad (104)$$

Taking $t_0 = T_k, t = T_{k+1-}$, the above becomes

$$\tilde{v}_k = \tilde{w}_k + \frac{1}{2}\left(1 - \exp\left(-2\sqrt{\frac{\mu}{\tilde{\kappa} R^2}}(T_{k+1} - T_k)\right)\right)(\tilde{z}_k - \tilde{w}_k) \quad (105)$$

$$\tilde{z}_{T_{k+1-}} = \tilde{z}_k + \frac{1}{2}\left(1 - \exp\left(-2\sqrt{\frac{\mu}{\tilde{\kappa} R^2}}(T_{k+1} - T_k)\right)\right)(\tilde{w}_k - \tilde{z}_k). \quad (106)$$

Comparing equation (105) and (14), we know $\tau_k = \frac{1}{2}\left(1 - \exp\left(-2\sqrt{\frac{\mu}{\tilde{\kappa}R^2}}(T_{k+1} - T_k)\right)\right)$ and $\tau_k'' = \frac{1}{2}\left(1 - \exp\left(-2\sqrt{\frac{\mu}{\tilde{\kappa}R^2}}(T_{k+1} - T_k)\right)\right)$. Furthermore, by using (105), we get

$$\tilde{w}_k - \tilde{z}_k = \frac{1}{1 - \frac{1}{2}\left(1 - \exp\left(-2\sqrt{\frac{\mu}{\tilde{\kappa}R^2}}(T_{k+1} - T_k)\right)\right)}(\tilde{v}_k - \tilde{z}_k). \tag{107}$$

Based on (106), (107), and (21), we conclude that $\tau_k' = \frac{1 - \exp\left(-2\sqrt{\frac{\mu}{\tilde{\kappa}R^2}}(T_{k+1} - T_k)\right)}{1 + \exp\left(-2\sqrt{\frac{\mu}{\tilde{\kappa}R^2}}(T_{k+1} - T_k)\right)}$. Moreover, we have $\tilde{\gamma}_k = \gamma_{T_k} = \frac{1}{R^2}$ and $\tilde{\gamma}_k' = \gamma_{T_k}' = \frac{1}{\sqrt{\mu\tilde{\kappa}R^2}}$. We can now conclude that the corresponding discrete-time algorithm under the above choice of parameters satisfies

$$\mathbb{E}\left[\exp\left(\sqrt{\frac{\mu}{\tilde{\kappa}R^2}}T_k\right)\frac{1}{2}\|\tilde{w}_k - w_*\|^2\right] \le \left(\frac{1}{2}\|\tilde{w}_0 - w_*\|^2 + \frac{\mu}{2}\|\tilde{z}_0 - w_*\|_{H(\tilde{w}_0)^{-1}}^2\right). \tag{108}$$

$\square$