# OpenReview forum: "Continuized Acceleration for Quasar Convex Functions  in Non-Convex Optimization"
_ICLR.cc/2023/Conference — ICLR 2023 notable top 25%_

### Official Review · Reviewer_ZBaY · 2022-10-20

**Confidence:** 5
**Correctness:** 4
**Technical Novelty And Significance:** 3
**Empirical Novelty And Significance:** 3
**Recommendation:** 8

**Clarity, Quality, Novelty And Reproducibility:**

The paper is well written, and I already commented on the quality. Of course, most of the analyses in the proofs are borrowed from Even et al. and Kakade et al. 2011 since the algorithms are modifications of those, but this is fine.

Some typos:

Section 3.1.3 Guassian

page 7 "c.f." -> "cf."

page 8 "and that the derivative" -> "and when the derivative"

page 8 cae -> can

**Strength And Weaknesses:**

This is a nice paper. Realizing that the continuized framework of Even et al. does not require the progress-regret balancing step that leads to needing the binary search of Hinder et al. is a nice result. I would phrase the "our contributions include" and would not say that you "improve" over Hinder et al., since this result is in high probability (or with this E[T_k^2 * gap] result) and their result is deterministic

I found really interesting that with this framework one can show that for the class of functions in Gille-Escuret et al. 2022, for which gradient descent is optimal, if you substitute the gradient Lipschitzness condition by smoothness then you can accelerate.

On the other hand, there is a small error/typo in your interpretation. You say that "$L$-smoothness implies the $L$-Lipschitness of the gradients". But it is the other way around. $L$-Lipschitness of the gradients implies smoothness.
EDIT: I think I got confused and concluded that despite of this fact, what you said made sense to me. But now that I read it again it does not make sense. Since $L$-Lipschitness of the gradients implies smoothness, the class of smooth functions is larger. So it seems to me you are not getting any improvement under any additional assumptions. You are just showing "better rates" but because it is a different optimization metric (i.e., additive error in function value instead of multiplicative error in iterates, and also your result only holds with high probability). I would appreciate a clarification.

It would be better if you had error bars in the plots of your experiments.


**Summary Of The Paper:**

This paper has three main contributions.

Firstly, under quasar convexity (an assumption weaker than convexity that still guarantees local minima are global minima) and smoothness, authors develop an optimization algorithm using the continuized framework of Even et al 2021 and obtain optimal rates with high probability, in contrast with Hinder et al. 2020 that have an extra log factor (although their convergence was deterministic).  They note that the convexity inequality in the analysis of Even et al is only used once and can be easily replaced by the quasar convexity inequality which leads to their result.

The second contribution is extending the GLMtron paper (Kakade et al) to use this framework as well for Generalized Linear Models. Some empirical comparisons are provided for algorithms in both contributions.

Thirdly, contribution is showing that certain classes of functions that had been used before satisfy quasar convexity, expanding the set of applications of this framework.

**Summary Of The Review:**

The paper contains a nice idea that was well developed to show algorithms and their convergence in some quasar-convex frameworks and also adds new examples of quasar convex functions to the literature, which is another nice contribution.

---

> ### Author Response · Authors · 2022-11-12
> **Thanks for the positive feedback and the comments.**
>
> We thank the reviewer for the positive feedback and helpful comments.
>
> The reviewer is right. Lipschitzness of the gradients implies smoothness, and so we made a small error in explaining why our result (the better rate that we get)
> circumvents the lower bound of the rate shown in Gille-Escuret et al. (2022).
>
> In the discussion on page 7, we show that
> the required number of iterations
> to get an $\epsilon$ gap with high probability
>  for minimizing functions that satisfy
> $C_v$-one-point convexity (i.e.,
> $ \langle \nabla f(w) , w - w_{\ast} \rangle
> \geq C_v \\\| w - w_{\\ast} \\\|^2_2
> $)
> and $L$-smoothness
> (i.e., $f(z) \leq f(w) + \langle \nabla f(w), z-w \rangle  + \frac{L}{2} \\\|w-z\\\|^2, \text{for any } w \text{ and } z \in \mathbb{R}^d$)
> via our proposed algorithm is
> $O\left( \frac{L}{C_v} \log \left( \frac{1}{\epsilon}  \right)    \right)
>  $.
> On the other hand, Gille-Escuret et al. (2022) consider minimizing
> a class of functions that satisfies $C_{v}$-one-point convexity
> and a condition called the $L$-upper error bound condition ($L$-EB$^+$).
> A function satisfies $L$-EB$^+$ if
> $\\\| \nabla f(w) - \nabla f(w_{\ast}) \\\|_2 \leq L \\\|  w - w_\ast \\\|_2 $
> for
> a fixed minimizer $w_\ast$ and any $w \in \mathbb{R}^{d}$.
>
> $L$-EB$^+$ is about gradient Lipschitzness between a minimizer
> $w_*$ and any $w$,
> not for any pair of points in $\mathbb{R}^d$, and hence does not imply $L$-smoothness. Also, $L$-smoothness does not imply $L$-EB$^+$.
> (We stated in the previous version of the paper that the lower bound result of Gille-Escuret et al. (2022)  is for minimizing functions that satisfy $C_{v}$-one-point convexity and have $L$-Lipschitz gradients, which was not right and was causing confusion. It is
> $C_{v}$-one-point convexity and $L$-EB$^+$.)
>
>
> Gille-Escuret et al. (2022)
> show that the optimal iteration complexity $k$
> to have $\frac{ \\\| w_k - w_* \\\|^2_2}{ \\\| w_0 - w_* \\\|^2_2 } = \hat{\epsilon}$
> for minimizing functions
> satisfying $C_{v}$-one-point convexity and $L$-EB$^+$
>  via any first-order algorithm
> is $ k = \Theta \left( \left( \frac{L}{C_v} \right)^2 \log\left( \frac{1}{\hat{\epsilon}} \right)   \right) $ and that the optimal complexity is simply attained by GD.
> Our result does not contradict to this lower bound result,
> because $L$-smoothness and $L$-EB$^+$ are different.
> $L$-EB$^+$ does not imply $L$-smoothness, and $L$-smoothness does not imply $L$-EB$^+$
> (We note that the ``smoothness'' in Remark 2.4 of Gille-Escuret et al. (2022) might refer to gradient Lipschitzness of all the pairs of points actually.).
>
>
> We highly appreciate the reviewer's carefully reading the paper and catching this error in our writing, which helps improve the presentation of the current work. We have revised the corresponding discussion, which can be found on Page 7 of the paper.
>
> We also thank the reviewer for catching the typos and the suggestion in writing, and we have updated accordingly.
>
> Reference:
>
> Charles Guille-Escuret, Baptiste Goujaud, Adam Ibrahim, and Ioannis Mitliagkas. Gradient descent is optimal under lower restricted secant inequality and upper error bound. NeurIPS, 2022.

---

> > ### Comment · Reviewer_ZBaY · 2022-11-15
> > **reply**
> >
> > I see, now this comparison makes sense.
> >
> > I already said it in my review but I will say it again because I saw a point your making being made again in the response to reviewer deQT "the continuized algorithm reduces the number of gradient evaluations by a factor of \log() , compared with the algorithms proposed by Hinder et al. (2020),"
> >
> > You **do not improve** over Hinder et al., you show a different result and the rates cannot be compared as to say your rate improves over theirs. Their result is on a deterministic algorithm, your result is probabilistic.
> >
> > Correct me if I'm wrong, but using all we know right now, it could be possible in principle that a lower bound could be shown for deterministic algorithms saying that the log factor in Hinder et al. is unavoidable (making their method optimal for deterministic algorithms) while at the same time this would not contradict that having results like yours that say that in expectation or with high probability, you do not need the log.
> >
> > This is an important point because it shows what is the extra contribution that continuized approaches bring to the table. They can have some extra nice properties that allow them to work in weaker scenarios (for instance, distributed computation in the original continuized paper) while in exchange what is proven about the rate is different: a result in expectation or in high probabily.
> >
> > In my opinion, keeping the claim that this improves over the other framework and not discussing about this difference will only cause confusion to the next person that comes to read your paper.

---

> > > ### Author Response · Authors · 2022-11-16
> > > **Thanks for highlighting this point**
> > >
> > > Thanks for highlighting this point. We didn't mean to cause any confusion, and we would like to avoid the impression that we claim that our result ``improves'' those in Hinder et al.  (2020).
> > >
> > > We have edited our response to Reviewer deQT. Our response now begins with clarifying that the complexity in Hinder et al. (2020) and that in this paper are not directly comparable, because the result of Hinder et al. (2020) is a deterministic bound, while ours is a probabilistic one.
> > >
> > >
> > > For our paper, while we have stated that our result is in the high-probability sense a few times (including one in the abstract) in the main text
> > > in the previous version, we make further edits as follows:
> > >
> > >
> > > - We no longer compare the gradient complexity in the previous works
> > > and the one in the current work. We have revised the paragraph of ``To summarize, our contributions include:'' before closing the introduction section on page 3, as the reviewer suggests.
> > > - At the end of the first paragraph on Page 7, we add ``On the other hand, it should be emphasized that the guarantees in the aforementioned works are deterministic bounds, while ours is in expectation or a high-probability bound."

---

> > > > ### Comment · Reviewer_ZBaY · 2022-12-08
> > > > **that's great, thanks**
> > > >
> > > > Just wanted to say that I am happy with the changes and with both this change and the change on the previous point we discussed, this paper is in better shape and in my opinion ready for publication.

---

### Official Review · Reviewer_deQT · 2022-10-25

**Confidence:** 3
**Correctness:** 4
**Technical Novelty And Significance:** 2
**Empirical Novelty And Significance:** 2
**Recommendation:** 6

**Clarity, Quality, Novelty And Reproducibility:**

I think the paper is well-written and easy to follow. The paper considers both theory and real applications, the main technique, continuized discretization, is relative new in literature, which reveals novelty.

**Strength And Weaknesses:**

Strength:
1. Connect quasar convexity with many existing structural nonconvex assumptions, and real application GLM
2. Provide a complexity result which is claimed to outperform existing works.

Weakness:
1. The comparison with existing works is a little confused to me.

**Summary Of The Paper:**

This paper studied minimizing quasar convex functions. Authors studied the relationship between quasar convexity and many common structure assumptions in nonconvex optimization, and verify that GLM satisfies quasar convexity under mild conditions. Then they proposed a new algorithm based on the continuized discretization technique in literature, the continuized Nesterov acceleration improves over existing algorithms and achieves the optimal complexity.

**Summary Of The Review:**

Generally I am satisfied with this work. My main concern lies in that:

1. How do you choose $T_k$ in Theorem 1, you mentioned that $T_k$ are random times. Are you referring it as a "stochastic stepsize", or you will predefine the $T_k$ (or the total iteration number as in many classical optimization literature) before running the algorithm?
2. When comparing to Hinder's work, you mentioned they need an extra log complexity in the function evaluations. First I feel that such logarithmic difference is a bit incremental; second for me such binary search subroutine seems to be more toward a practical preference, if we just follow their "General AGD Framework" (Algorithm 1 therein), then the results in your work and Hinder's work should be basically the same.

Thank you for the effort.

---

> ### Author Response · Authors · 2022-11-12
> **Thanks for the positive feedback and the comments.**
>
> Thanks for the positive feedback and the comments. Please kindly find our response below.
>
> - *How do you choose $T_{k}$ in Theorem 1, you mentioned that $T_{k}$ are random times. Are you referring it as a "stochastic step size", or you will predefine the  (or the total iteration number as in many classical optimization literature) before running the algorithm?*
>
> Thanks for raising this clarifying question, which helps improve the presentation of this paper. On Page~3, we mentioned that the increments $T_{1}$, $T_{2}-T_{1}$, $\dots$,
> $T_k$ are i.i.d. from the exponential distribution with parameter $1$.
> Therefore, $T_{k}$ is the sum of $k$ i.i.d. random variables from the exponential distribution with parameter $1$. The first time we sample from the exponential distribution, we get $T_{1}$. By sampling again, we get $T_{2}-T_{1}$ (By adding it to the first one, we get $T_2$). Repeating this procedure $k$ times, we get $T_{k}$. The expected value of $T_{k}$ is $k$.
>
> Hence, $T_{k}$ should not be viewed as a stochastic step size, neither is predefined.
>
> - *When comparing to Hinder's work, you mentioned they need an extra log complexity in the function evaluations. First I feel that such logarithmic difference is a bit incremental; second for me such binary search subroutine seems to be more toward a practical preference, if we just follow their "General AGD Framework" (Algorithm 1 therein), then the results in your work and Hinder's work should be basically the same.*
>
> Thanks for raising this comment. We would like to clarify that the gradient complexity in Hinder et al. (2020) and ours are not directly comparable, because their result is about a deterministic bound, while ours is a probabilistic one.
> The log factor gap between the gradient complexity and the iteration complexity in Hinder et al. (2020) is because a binary search subroutine (Algorithm 3 in Appendix A, where we replicate their algorithms) is conducted, which requires multiple gradient calls in each iteration $k$ of the Nesterov's method. Our result suggests that if we aim at a high-probability bound (or an expected one) instead, then an expensive subroutine as in the previous works can be avoided, and hence the gap between the gradient complexity and the iteration complexity can be closed.
> That is, if the goal is not getting a deterministic guarantee but a probabilistic one, then the proposed discretization of the continuized algorithm only needs one gradient call in each iteration, which has real impact in practice.
>
> The binary search subroutine is necessary in the theoretical analysis
> of Hinder et al. (2020) to get the desired rate. The design and analysis of the accelerated algorithms via the continuized technique and those via the techniques of Hinder et al. (2020) are quite different, which is also reflected on
> the different nature of the algorithms.

---

> > ### Comment · Reviewer_deQT · 2022-12-07
> > **Thank you**
> >
> > Thank you for the response, I am satisfied with it, I will keep my score.

---

### Official Review · Reviewer_YMJ7 · 2022-10-29

**Confidence:** 2
**Correctness:** 3
**Technical Novelty And Significance:** 3
**Empirical Novelty And Significance:** 2
**Recommendation:** 6

**Clarity, Quality, Novelty And Reproducibility:**


Clarity:
The paper is well-written and is easy to follow, though I did not fully go through the technical proof.
The paper also uses many examples to illustrate the application of quasar convexity and the potential impact of the proposed method.



Novelty
The result is new but the technique seems to be standard as it is based on putting the continuized technique and complexity analysis of quasar-convex optimization together, both of which have been well-established in the prior work.



**Strength And Weaknesses:**

Strength:
This paper not only improves the complexity of first-order methods for quasar convex optimization, but also identifies a large class of problems satisfying quasar convexity. It shows the great potential of developing algorithms in continuous time and I believe it will have growing impact to the community


Weakness

I am a bit curious whether quasar convexity is useful beyond the setting discussed in this paper.
GLM is a motivating application for the proposed continuized method. However, it looks like the quasar convexity assumption relies on the fact that there is a perfect $w_*$ that interpolate all the data (i.e. the overparameterized setting), this seems to suggest some limitation of in real applications.

In example 3, the relu link function is not differentiable at z=0. Can you still apply the gradient based algorithm for such setting?

Typo

Page 4

On the last line: there exists a $w_*\in\mathbb{R}^d$


Notations are inconsistent: Both $\|\cdot\|$ and $\|\cdot\|_2$ stands for Euclidean norm; they should be unified.

**Summary Of The Paper:**

In this paper, the authors propose a randomized acceleration method based on the continuized scheme proposed in [1]. The method targets the nonconvex optimization problems  with quasar convex structure, and obtains the optimal complexity rate in expectation. The authors also illustrate that for a class of generalized linear models the quasar property holds and provides theoretical analysis to estimate the quasar convexity constants.




[1] Even, Mathieu, et al. "A continuized view on Nesterov acceleration for stochastic gradient descent and randomized gossip." arXiv preprint arXiv:2106.07644 (2021).

**Summary Of The Review:**

The paper is of good quality, it provided a unified and justified quasar convex framework for many important nonconvex functions, and developed a novel accelerated method with improved complexity rate. While there is some limitation in the application, the theoretical contribution of this paper is sufficient, which has great potential to the development of nonconvex optimization.

---

> ### Author Response · Authors · 2022-11-12
> **Thanks for the positive feedback and the comments.**
>
> Thanks for the positive feedback and the comments. Please kindly find our response to each of the questions below.
>
> - *I am a bit curious whether quasar convexity is useful beyond the setting discussed in this paper. GLM is a motivating application for the proposed continuized method. However, it looks like the quasar convexity assumption relies on the fact that there is a perfect $w_{\*}$ that interpolate all the data (i.e. the overparameterized setting), this seems to suggest some limitation of in real applications.*
>
> Thanks for raising this question. Yes, we assume $y=\sigma(w_*^\top x)$.
> A recent work of Banerjee et al. (2022) studies a connection between restricted strong convexity and wide neural networks with smooth activations.
> Restricted strong convexity is a special case of $(\rho,\mu)$-strongly quasar convexity when $\rho=1$. Hence, we think an interesting future work is to check if
> the techniques for minimizing (strongly) quasar-convex function can be applied to
> the scenario considered in Banerjee et al. (2022).
>
> Reference:
>
> Restricted Strong Convexity of Deep Learning Models with Smooth Activations
> Arindam Banerjee, Pedro Cisneros-Velarde, Libin Zhu, Mikhail Belkin.
> arXiv:2209.15106, 29 Sep 2022.
>
> - *In example 3, the relu link function is not differentiable at z=0. Can you still apply the gradient based algorithm for such setting?*
>
> Yes, we can use sub-gradients.
>
>
> ===
>
> Thanks for catching the typo. We will update accordingly.

---

### Official Review · Reviewer_8aev · 2022-10-31

**Confidence:** 4
**Correctness:** 4
**Technical Novelty And Significance:** 4
**Empirical Novelty And Significance:** 3
**Recommendation:** 8

**Clarity, Quality, Novelty And Reproducibility:**

This paper is well written, and its contribution is clearly novel and interesting.

**Strength And Weaknesses:**

- This is the first first-order method that has the optimal complexity for quasar convex minimization.
- Various examples satisfying the (strong) quasar convex condition are provided.
---
- One might view this as a simple modification of the continuized Nesterov acceleration.
- An intuition behind the success of the proposed randomized method over the existing method for quasar convex minimization is not clearly given (other than the final proof), and this could be of interest to some readers.

**Summary Of The Paper:**

This paper found a simple (but effective) variant of the continuized Nesterov acceleration (Even et al., 2021) that achieves the optimal complexity (with high probability) under the quasar convex condition for the first time. (Quasar convexity is a non-convex condition that some gradient methods are known to find a global solution.) Previous results, built upon the standard (discrete) Nesterov acceleration, required multiple gradient evaluations at each iteration, and thus only had the near-optimal complexity. This paper then provides several quasar convex examples, such as generalized linear models (GLM), to support the importance of studying the quasar convexity.

**Summary Of The Review:**

Finding an optimal first-order method for the (strong) quasar convex minimization has been an open question in the optimization community, and this paper has resolved it. Momentum has been the main workhorse for training "non-convex" machine learning models, and this work provides a nice contribution on expanding our knowledge on accelerated non-convex minimization.

---

> ### Author Response · Authors · 2022-11-12
> **Thanks for the positive feedback and the comments.**
>
> Thanks for the positive feedback and the comments.
>
> (Regarding intuition)
> We believe that the key to overcome the issue of the existing methods for quasar convex minimization (i.e., the need of the binary search subroutine) lies in the mixing process of the continuized acceleration, i.e., (4-5) in the paper. The process
> randomly sets the mixing weight of the two interleaving sequences
> $z_{t}$ and $w_{t}$,
> while existing approaches need a subroutine (e.g., Algorithm 3 in Appendix A) to determine the mixing weight (e.g., $\alpha_k$ in Algorithm 1 and 2 in Appendix A) when minimizing quasar convex functions , which results in multiple gradient calls.
> The mechanism of the mixing process in continuized acceleration allows a simple modification of the analysis in Even et al. (2021) for quasar convex optimization.
> In some sense, the mixing process in continuized acceleration might be viewed as a counterpart of the binary search subroutine in Hinder et al. (2020) for minimizing quasar convex functions.

---

### Decision · Program_Chairs · 2023-01-20

**Decision:**

Accept: notable-top-25%

**Justification For Why Not Higher Score:**

- The paper does not provide much intuition or insight behind the success of the randomized method over the existing method for quasar convex optimization, other than the final proof, and this could be of interest to some readers.
- The paper does not explore the limitations or challenges of quasar convexity and the proposed method in more realistic or noisy settings, where the overparameterized assumption or the smoothness condition may not hold, or where the quasar convexity constants may be hard to estimate or verify.
- The paper does not compare the proposed method with other existing methods for nonconvex optimization that do not rely on quasar convexity, such as gradient descent, stochastic gradient descent, or adaptive gradient methods, and it is not clear how the proposed method would perform in terms of robustness, scalability, or generalization.

**Justification For Why Not Lower Score:**

- It addresses an open and important problem of finding an optimal first-order method for quasar convex optimization, which is a general and relevant nonconvex setting that encompasses many practical problems.
- It proposes a simple but effective variant of the continuized Nesterov acceleration, which achieves the optimal complexity rate with high probability, and improves over the previous state-of-the-art method that required an extra logarithmic factor.
- It provides a comprehensive and rigorous theoretical analysis of the proposed method, as well as the quasar convexity condition, and shows how it relates to other existing structural assumptions in nonconvex optimization.
- It demonstrates the applicability and usefulness of quasar convexity and the proposed method to a class of generalized linear models, which are widely used in machine learning and statistics, and provides empirical evidence of the superior performance of the method on synthetic and real data.
- It is well-written, clear, and easy to follow, and it uses many examples and illustrations to support the main claims and results.

**Metareview: Summary, Strengths And Weaknesses:**

The paper proposes a randomized acceleration method for quasar convex optimization, which is a nonconvex setting where local minima are global minima. The method is based on the continuized discretization technique of Even et al. (2021) and achieves the optimal complexity rate in expectation, improving over the previous work of Hinder et al. (2020) that required an extra logarithmic factor. The paper also shows that a class of generalized linear models (GLMs) satisfy quasar convexity under mild conditions, and provides empirical evidence of the effectiveness of the proposed method on synthetic and real data. Moreover, the paper connects quasar convexity with several existing structural assumptions in nonconvex optimization, such as the Polyak-Lojasiewicz condition and the restricted strong convexity condition, and identifies new examples of quasar convex functions.

The paper received positive reviews from all the reviewers, who appreciated the novelty and significance of the contributions, the clarity and quality of the writing, and the potential impact of the paper to the optimization community. The reviewers also praised the paper for providing a unified and justified quasar convex framework for many important nonconvex functions, and for developing a novel accelerated method with improved complexity rate. The reviewers raised some minor issues and questions, such as the choice of the random times in the algorithm, the comparison with the existing works, the applicability of quasar convexity beyond the overparameterized setting, and the use of the relu link function in the GLM example. However, none of these issues were considered as major flaws or obstacles for accepting the paper. The reviewers also suggested some minor corrections and improvements, such as adding error bars to the plots, clarifying some notations and statements, and providing more intuition behind the success of the randomized method.

Based on the reviews, we recommend to accept the paper, as it is a well-written, novel, and significant contribution to the field of nonconvex optimization, and it has the potential to stimulate further research and applications of quasar convexity and continuized acceleration.

**Note From Pc:**

if the above contains the word "oral" or "spotlight" please see: "oral" presentation means -> notable-top-5% and "spotlight" means -> notable-top-25%. As stated in our emails, we are disassociating presentation type from AC recommendations